# Bidirectional and parallel relationships in macaque face circuit revealed by fMRI and causal pharmacological inactivation

Ning Liu ®[1,2] ✉, Marlene Behrmann[3,4], Janita N. Turchi[5], Galia Avidan ®[6], Fadila Hadj-Bouziane ®[1,7,8] & Leslie G. Ungerleider[1]

Although the presence of face patches in primate inferotemporal (IT) cortex is well established, the functional and causal relationships among these patches remain elusive. In two monkeys, muscimol was infused sequentially into each patch or pair of patches to assess their respective influence on the remaining IT face network and the amygdala, as determined using fMRI. The results revealed that anterior face patches required input from middle face patches for their responses to both faces and objects, while the face selectivity in middle face patches arose, in part, from top-down input from anterior face patches. Moreover, we uncovered a parallel fundal-lateral functional organization in the IT face network, supporting dual routes (dorsal-ventral) in face processing within IT cortex as well as between IT cortex and the amygdala. Our findings of the causal relationship among the face patches demonstrate that the IT face circuit is organized into multiple functional compartments.

Faces convey a wealth of information (e.g., identity, emotion, and personality traits) critical for our daily social interactions. Given the importance of faces, it is unsurprising that neuroimaging studies conducted in humans and monkeys have revealed a set of "face-selective" cortical regions (or "face patches") within the infer-otemporal (IT) cortex, which respond more strongly to faces compared to non-face objects[1–3]. Understanding the signal propagation and circuitry among these face patches would greatly improve our understanding of the neural mechanisms underlying face processing, and potentially extend this knowledge to our understanding of object processing, as well.

Several previous studies have demonstrated that, at the anatomical level, the face patches are interconnected into a face-processing network (for review, see refs. [4, 5]). For example, combined electrical microstimulation with simultaneous fMRI revealed that stimulation of specific face patches produced strong activation in a subset of the other face patches[6]. Furthermore, injecting retrograde tracers into specific face patches demonstrated that these face patches share dense feedforward and feedback anatomical connections[7]. While these previous studies have confirmed that face patches are organized into a specific network for face processing, the functional organization of this network still remains elusive.

It is generally agreed that object representations become more complex as one moves anteriorly along the visual pathway[8]. Such a posterior-anterior hierarchical organization in the face-processing network has been supported by functional evidence: neurons in the more anterior face patches respond more invariantly in view, size, and position than in the posterior patches[9]. Unsurprisingly, given that the pathways in the visual cortex are commonly bidirectional[8], dense feedback projections also influence the interactivity within the face processing network in IT cortex. For example, in the middle face pat-ches [in the medial fundal (MF) and medial lateral (ML) portions of IT

[1]Section on Neurocircuitry, Laboratory of Brain and Cognition, NIMH, NIH, Bethesda, MD 20892, USA. [2]State Key Laboratory of Brain and Cognitive Science, Institute of Biophysics, Chinese Academy of Sciences, 100101 Beijing, China. [3]Department of Ophthalmology, University of Pittsburgh, Pittsburgh, PA 15213, USA. [4]Department of Psychology and Neuroscience Institute, Carnegie Mellon University, Pittsburgh, PA 15213, USA. [5]Laboratory of Neuropsychology, NIMH, NIH, Bethesda, MD 20892, USA. [6]Department of Psychology, Ben-Gurion University of the Negev, Beer-Sheva 8410501, Israel. [7]INSERM, U1028, CNRS UMR5292, Lyon Neuroscience Research Center, ImpAct Team, F-69000 Lyon, France. [8]University UCBL Lyon 1, F-69000 Lyon, France. ✉ e-mail: liuning@ibp.ac.cn

cortex], single-unit recordings have revealed a face-selective response characterized by two peaks, occurring respectively at 130 ms and 200 ms[10], with the latter likely reflecting feedback from more anterior face patches. Moreover, it has been found that the prediction errors (deviation of actual from predicted stimuli) in ML do not reflect its own tuning but those of the anterior patch (AL/AM), implicating feedback projections[11]. Relatedly, computational vision models of face processing that incorporate this bidirectional organization qualitatively and quantitatively capture the tuning properties of the middle face patches to facial features[12]. However, the potential bidirectional organization within the face network at the functional level has not been directly addressed nor elucidated. Furthermore, the nature of information being transferred in the bidirectional circuit and the causal relationships among the face patches remain to be determined.

Beyond the antero-posterior bidirectional route, in humans, there exist two pathways of face processing: a ventral pathway mediating the invariant aspects of the face and a dorsal pathway mediating the dynamic properties of the face[13,14]. Functional evidence has shown that, in monkeys, as in humans, fundal, but not lateral, patches are predominantly involved in processing dynamic facial information (e.g., motion and emotional expression)[15–17]. Thus, it is possible that there exists a fundal-lateral axis in the face network in monkeys but, to date, the existence of such parallel functional organization has not been clearly documented[6,7]. Of note, face patches within the superior temporal sulcus (STS) fundus have been less targeted in previous studies, and this might have hampered the ability to uncover a potential fundal-lateral axis in the IT face network in monkeys.

Finally, the amygdala is one of three subcortical brain structures that consistently exhibits strong anatomical and functional connections with IT face patches[6,7,18] and represents information relevant for emotional and social perception[19,20]. Face responsive/selective cells and voxels have also been found in the amygdala[21–23] and stimulating face patches (e.g., AM) evokes activation of the amygdala[6]. As might be predicted, lesions of the amygdala selectively eliminate responses to emotional faces while preserving the functional integrity of face patches within IT cortex[18]. Recent evidence in humans has revealed a selective functional pathway projecting along the STS to the amygdala subserving the processing of dynamic face information[24,25]. Tracer studies in macaques have also identified a pathway along the STS to the lateral nucleus of the dorsal amygdala[26]. However, it remains uncertain whether, if the fundal-lateral axis exists in the IT face network in monkeys, the functional information to the amygdala is funneled mainly through the fundal rather than the lateral pathway, as in humans. Therefore, in the present study, we also examined the functional relationship of the amygdala to the IT face network.

The studies of connectivity in the face network thus far have primarily been correlational in nature and causal conclusions have not been drawn. It is, however, crucial to understand the functional organization within the face-processing network. The approach of transient or reversible inactivation has proved to be a powerful tool in exploring the functional roles of specific brain areas and providing critical causal insights, with the advantage of being repeatable, with interleaved recovery periods. This approach has been broadly used in behavioral studies, combined with electrophysiological recordings and, more recently, with fMRI in macaques. Combining reversible inactivation with fMRI in particular, permits the investigation of whole-brain activity and provides a unique opportunity to assess the causal relationship within brain networks (for review, see ref. [27]).

Here, we aimed to characterize the functional and causal relationships among the IT face patches. We used fMRI combined with pharmacological inactivation (using muscimol, a GABA$_A$ agonist) of specific face patches within the IT cortex to clarify the relative contribution of the different patches and elucidate their functional organization along two axes: antero-posterior and fundal-lateral. We compared the responses of face patches to images of faces and

objects following muscimol versus control injections in individual or pairs of patches. We also examined the responses of the amygdala following inactivation of face patches to explore how face processing information is transmitted beyond the IT cortex, predicting that disrupting face patches within the fundus of STS might impact the amygdala response to a greater degree than inactivating the lateral face patches.

## Results

We injected muscimol into four face patches singly and in pairs in each hemisphere, spanning the posterior-anterior extent and fundal-lateral extent of the temporal lobe of two monkeys: MF, ML, AF and AL. Temporal face patches, identified using an independent localizer (Fig. 1A), are shown on both hemispheres of monkeys C and D on a lateral view of the inflated cortex in Fig. 1B, C. The results are divided into two sections. First, we systematically examine the pairwise effects of inactivation of a single or two face patches on the other patches and on the amygdala, relative to a control condition, and compare the magnitude of the effects within each hemisphere when monkeys viewed faces or viewed objects. We sequentially present the data from the inactivation of the middle face patches first and then from the inactivation of the anterior face patches. Second, we present analyses that permit inferences about the causal relationship between the different face patches and the amygdala. The results from individual animals and hemispheres showed slight variations from the common pattern which is described below (for additional details, see Figs. S1–10). We also include data from non-face patches to indicate the specificity of the key findings to the face patches alone. Note that, given the lack of consistency and small sample size, we refrain from discussing further any left versus right hemispheric differences in the present study.

### Impact of inactivation of the middle face patches

Figures 2 and 3 show the responses to faces (top panels) and objects (bottom panels) across the four face patches of the two monkeys following either control or muscimol injections in the middle face patches. The GLMMs analysis revealed significant main effects of Treatment (4 levels: Control, combined F and L inactivation, F inactivation, and L inactivation) and Hemisphere (2 levels: ipsilateral and contralateral hemisphere to the injection sites), which were largely qualified by interactions between the two factors (all the results are reported in Table S1). These findings indicate that inactivation of the middle face patches resulted in differential responses in the hemispheres ipsilateral and contralateral to the inactivation sites. This was true for responses to faces in all four face patches (MF, ML, AF, and AL) and the amygdala (all $p$ values < 0.001), and this was also found for responses to objects in three out of 4 patches (MF, ML, and AF but not AL) with a marginal effect in the amygdala ($p = 0.053$).

Post hoc tests further revealed that, in the ipsilateral hemisphere, compared with data obtained from the control sessions (vehicle infusion in the same locations), muscimol infusions targeting both MF&ML significantly ($p < 0.001$) eliminated responses to both faces (Fig. 2A) and non-face objects (Fig. 2B) in MF and ML, confirming successful inactivation. Importantly, we also found a remote effect in the anterior patches (AF and AL), with significantly reduced responses to both faces ($p < 0.001$) and non-face objects (AF: $p < 0.01$, AL: $p = 0.009$ uncorrected). The remote changes in the anterior face patches indicate that responses to faces and objects in the anterior face patches require feedforward inputs from the middle face patches.

As anticipated, the impact of inactivating either MF or ML separately was more restricted (Fig. 2). Compared with data obtained from the control sessions, successful inactivation of MF alone eliminated its own responses (faces: $p < 0.001$; objects: $p < 0.001$) and significantly reduced responses to faces remotely in AF and ML ($p < 0.001$), but not

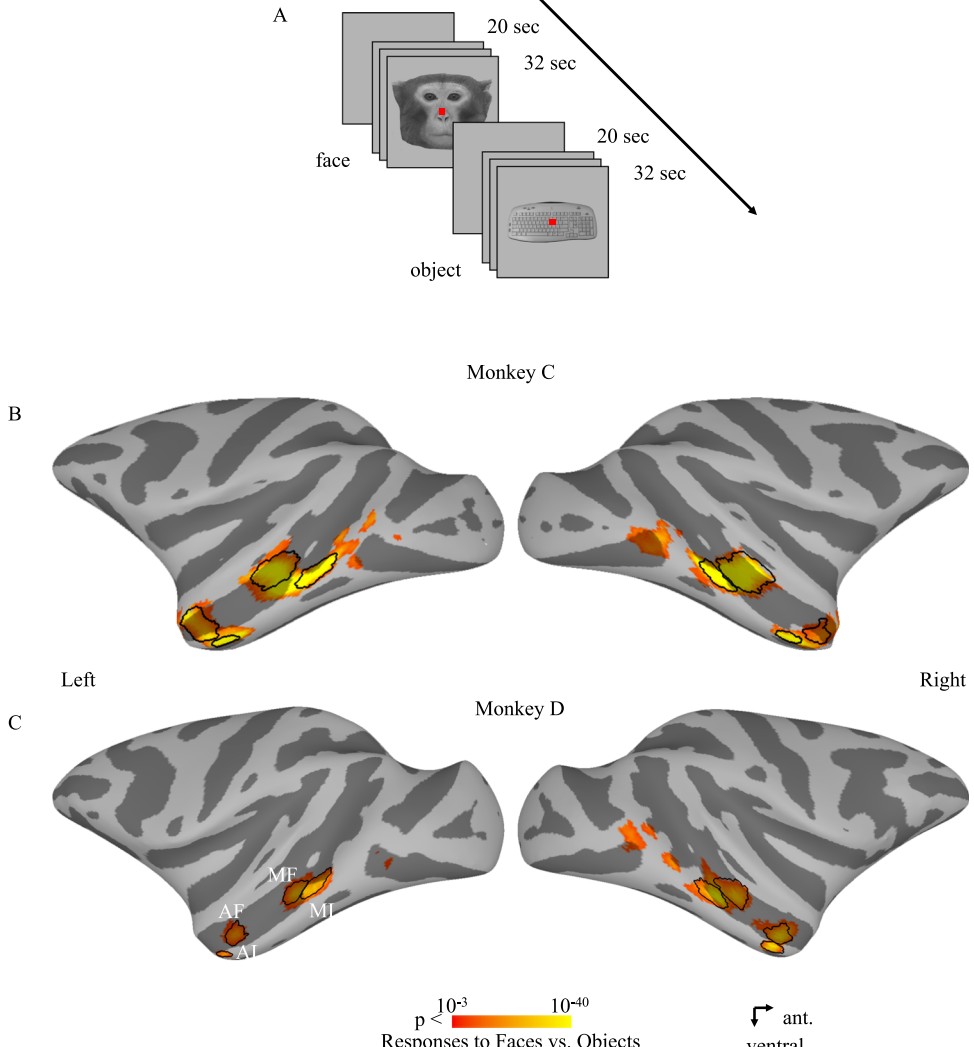

**Fig. 1 | The fMRI experiment design and face-selective patches in Monkeys C and D. A** An illustrative example of the fMRI experiment design. **B**, **C** Face-selective (neutral monkey faces versus familiar objects, $p < 0.005$ uncorrected) activation maps from the initial localizer sessions are shown on lateral views of the inflated cortex. The main selected voxels in each ROI are encircled by black lines. MF middle fundal face patch, ML middle lateral face patch, AF anterior fundal face patch, AL anterior lateral face patch.

AL, as well as responses to objects in AF ($p < 0.001$) but not ML or AL. Likewise, inactivation of ML alone was also successful (faces: $p < 0.001$; objects: $p < 0.001$) and significantly reduced responses to faces remotely in all the other face patches ($p < 0.001$) as well as responses to objects in MF ($p < 0.001$) but not in AF or AL.

Interestingly, as examined using Pearson's correlations (2-tailed), we found significant correlations in the responses to faces between anterior and middle face patches within the fundus of STS when inactivating MF (with AF: $r = 0.999$, $p = 0.001$) but not between face patches on the lateral surface of the IT cortex (AL: $r = 0.933$, $p = 0.067$ or ML: $r = 0.840$, $p = 0.160$). Conversely, there were significant correlations in the responses to faces between anterior and middle face patches on the lateral surface of the IT cortex when inactivating ML (with AL: $r = 0.959$, $p = 0.049$) but not between face patches within the fundus of STS (AF: $r = 0.826$, $p = 0.174$; MF: $r = 0.840$, $p = 0.160$) (Fig. 2A). These results indicate a functional organization based on parallel processing along the fundal-lateral axis.

Note that the beta coefficients in the inactivation sites were not zero (although probably not significantly different from zero, see Figs. S11 and 12A), which might result from the fact that there was not 100% coverage of targeted ROIs for inactivation (Tables S2 and 3). The overlap volume between the muscimol injection (measured with Gd-

signal) and targeted ROIs covered 49.48% of voxels within the targeted ROIs (MF: 55.14%; ML:45.71%) on average (theoretical overlap in MF: 61.84%; theoretical overlap in ML: 53.81%), suggesting that a large fraction of the middle face patches was silenced. We present the results on lateral views of the inflated cortex to provide a complete picture of the effects of inactivation. As shown in Figs. S11 and 12A, there were no, or only a few, voxels that survived the selected threshold ($p < 0.005$ uncorrected) after inactivation (especially within the defined ROIs).

Finally, compared to the effect within the ipsilateral hemisphere, the impact of face patches inactivation on the contralateral hemisphere was rather minimal and did not follow a consistent pattern, highlighting the dominance of ipsilateral processing at the level of IT cortex (Figs. 3, S1B, and S3–6B).

To explore the functional relationship between the face patches and the amygdala, we also investigated the impact of inactivation of the middle face patches on the amygdala. We found that inactivations of the face patches modulated responses to faces and objects in the amygdala ipsilateral to the inactivation site (Fig. S13A).

In the temporal cortex, face patches were surrounded by non-face-selective regions (e.g., object-selective regions). Therefore, to understand whether and how the face network connects with

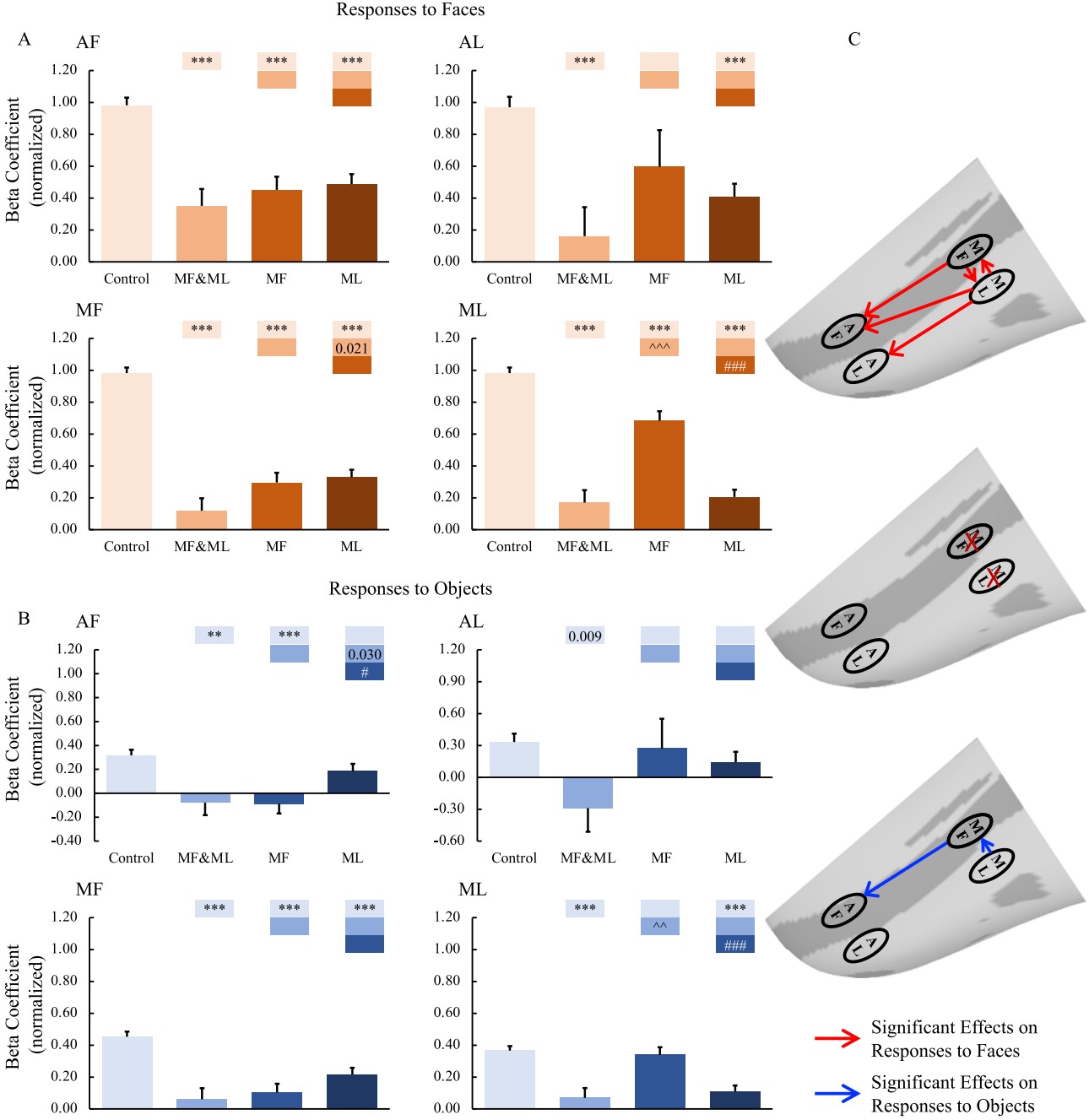

**Fig. 2 | Effects of middle face patch inactivation on responses in the ipsilateral temporal face patches. A, B** Effects of middle face patch inactivation on responses to faces and objects in the temporal face patches in the hemisphere ipsilateral to the inactivation sites across both monkeys. Box plots display mean values ± SEM. The name of the ROI is shown at the top left of each panel. MF&ML: combined MF and ML inactivation; MF: MF alone inactivation; ML: ML alone inactivation. Generalized Linear Mixed Models were performed for each ROI (AF, MF, and ML: $n = 664$; AL: $n = 566$) with Treatment and Hemisphere as fixed factors, and Monkey (C and D), L-R hemisphere (Left and Right), Session, and Run as random factors. Post hoc testing was done with correction for multiple testing using the Holm-Bonferroni method. Color bars above a particular histogram contain the statistical results of comparing this inactivation condition and the condition to its left with the same color as the bar: *, significant difference from control (**$p < 0.01$, ***$p < 0.001$, in light red/blue boxes); ^, significant difference from combined MF and ML inactivation (^^$p < 0.01$, ^^^$p < 0.001$, in middle red/blue boxes); #, significant difference from MF alone inactivation (# $p < 0.05$, ### $p < 0.001$, in dark red/blue boxes); value, uncorrected $p$ value; empty bar, no significant difference no matter whether multiple comparison corrections were done or not. **C** The schematic diagram of the effects of inactivation of the middle face patches. The top and bottom panels show the significant effects of middle face patch inactivation on responses to faces (red lines) and objects (blue lines) in the ipsilateral hemisphere, respectively. The middle diagram shows the location of the inactivation (marked with red X). MF middle fundal face patch, ML middle lateral face patch, AF anterior fundal face patch, AL anterior lateral face patch.

surrounding non-face-selective cortex, it is also important to investigate whether the inactivation of face patches might affect the object-selective regions. In the present study, we did not find any significant influence of inactivation of the middle face patches on the object-selective regions (Figs. S11 and 12A), consistent with previous micro-stimulation as well as tracer studies and also indicating that the remote effect of inactivation found in the present study were not directly caused by the muscimol spread.

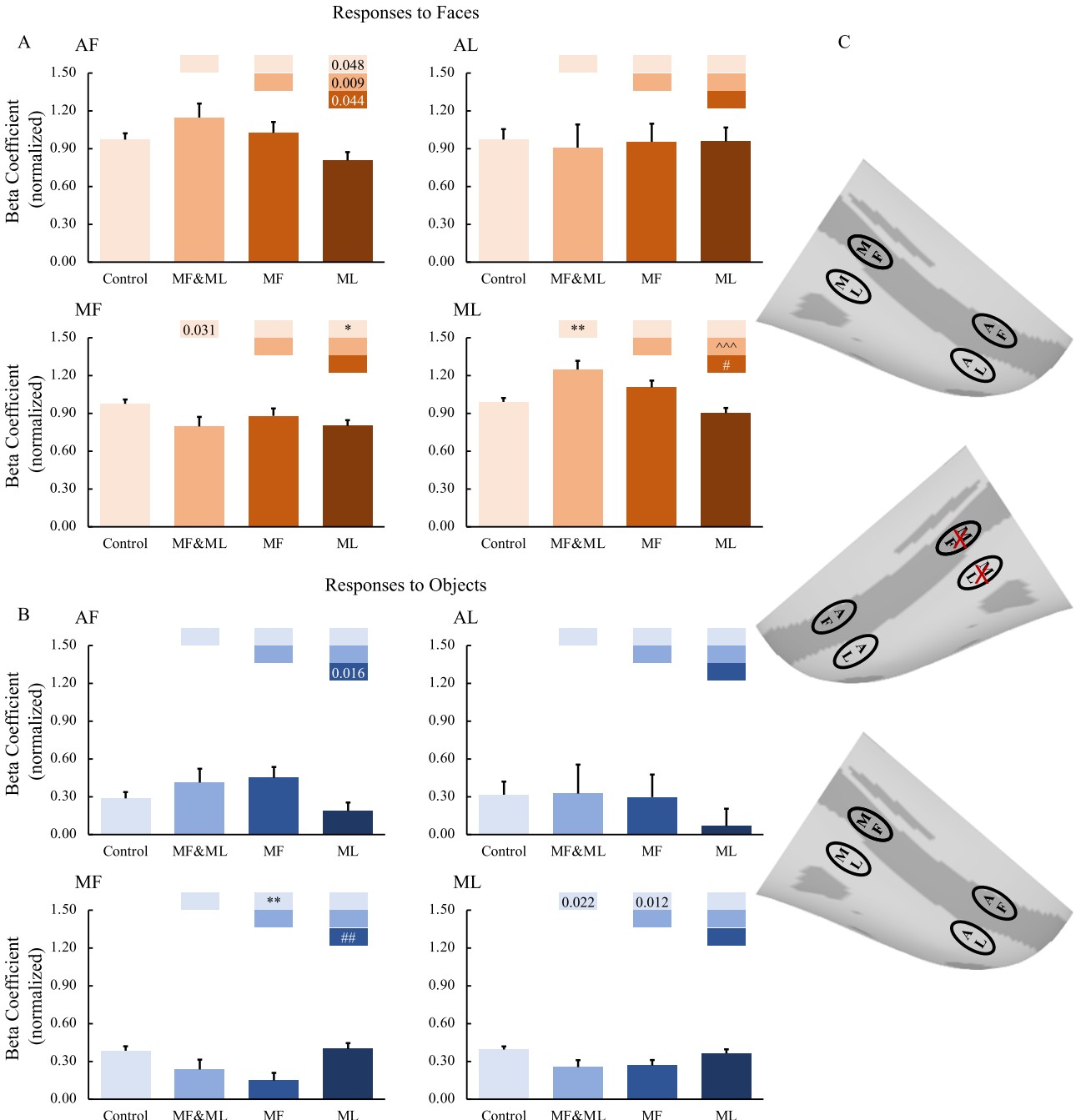

**Fig. 3 | Effects of middle face patch inactivation on responses in the contralateral temporal face patches. A, B** Effects of middle face patch inactivation on responses to faces and objects in the temporal face patches in the hemisphere contralateral to the inactivation sites across both monkeys. Box plots display mean values ± SEM. The name of the ROI is shown at the top left of each panel. MF&ML: combined MF and ML inactivation; MF: MF alone inactivation; ML: ML alone inactivation. Generalized Linear Mixed Models were performed for each ROI (AF, MF, and ML: n = 664; AL: n = 566) with Treatment and Hemisphere as fixed factors, and Monkey (C and D), L-R hemisphere (Left and Right), Session, and Run as random factors. Post hoc testing was done with correction for multiple testing using the Holm-Bonferroni method. Color bars above a particular histogram contain the statistical results of comparing this inactivation condition and the condition to its left with the same color as the bar: *, significant difference from control (*p < 0.05, **p < 0.01, in light red/blue boxes); ^, significant difference from combined MF and ML inactivation (^^^p < 0.001, in middle red/blue boxes); #, significant difference from MF alone inactivation (#p < 0.05, ##p < 0.01, in dark red/blue boxes); value, uncorrected p value; empty bar, no significant difference no matter whether multiple comparison corrections were done or not. **C** The schematic diagram of the effects of inactivation of the middle face patches. The top and bottom panels show no significant effects of middle face patch inactivation on responses to faces and objects in the contralateral hemisphere, respectively. The middle diagram shows the location of the inactivation (marked with red X). MF middle fundal face patch, ML middle lateral face patch, AF anterior fundal face patch, AL anterior lateral face patch.

## Impact of inactivations of the anterior face patches

The above results indicate that responses to faces and objects in the anterior face patches require feedforward inputs from the middle face patches. The obvious next question is whether there exist feedback projections from the anterior face patches to the middle ones, and if so, what roles these projections play. For this aim, we performed inactivations in the anterior face patches and then investigated the responses to both faces and objects in the middle face patches. The

same GLMMs on responses to faces/objects in each ROI separately as described above were conducted (Table S4). We found significant interactions between Treatment and Hemisphere for responses to faces in all 4 face patches as well as for responses to objects in ML, AL, and in AMG but not in MF or AF. Again, these findings indicated the differential influence of inactivation of the anterior face patches on face patches in the ipsilateral and contralateral hemispheres to the inactivation sites.

First, we examined the effects of combined AF and AL inactivation. The muscimol infusion eliminated responses to both faces (Fig. 4A, $p < 0.001$) and non-face objects (Fig. 4B, AF: $p < 0.05$, AL: $p = 0.014$ uncorrected) in AF and AL, confirming the success of the AF and AL inactivation. Responses in middle face patches (MF and ML) were also reduced but only in response to faces ($p < 0.001$) and not to non-face objects. These findings indicate that the face selectivity in the middle face patches arises, in part, from top-down inputs from the anterior face patches. To confirm these findings, face-selective indexes were calculated (Eq. 1). As shown in Fig. 5, the face-selective indexes in MF and ML were significantly modulated by combined AF and AL inactivation ($p < 0.01$).

$$Face\ Selectivity\ Index = \frac{\mu_{\text{faces}} - \mu_{\text{objects}}}{\sqrt{(\sigma^2_{\text{faces}} + \sigma^2_{\text{ojects}})/2}} \tag{1}$$

Next, we targeted AF and AL individually. Inactivation of AF alone (faces: $p < 0.001$, objects: $p = 0.045$ uncorrected) as well as inactivation of AL alone (faces: $p < 0.001$, object: n.s.) significantly reduced responses to faces in all the other face patches (AF alone inactivation: $p < 0.001$; AL alone inactivation: AF: $p < 0.001$, MF: $p = 0.015$ uncorrected, ML: $p < 0.01$).

Note that, in contrast with combined AF and AL inactivation, inactivation of AF or AL alone (especially AL) also reduced responses to objects in MF and ML (AF inactivation, MF: $p = 0.014$ uncorrected, ML: $p = 0.016$ uncorrected; AL inactivation, MF: $p < 0.001$, ML $p < 0.001$). The number of runs under inactivation of AF ($n = 43$) or AL ($n = 26$) alone was fewer than that under combined AF and AL inactivation ($n = 68$) (Table S5). Moreover, when we looked at individual hemisphere data (Figs. S2A and 7–10A), it appears that this result might mainly be the product of the left hemisphere of Monkey D (Fig. S9), with these data having been collected at the very end of the experiments. Due to the MION accumulation, the signals were much weaker than data collected earlier in the experiments, potentially contributing to the reduction in responses to objects after inactivation of AF or AL alone.

Note that inactivation of the anterior face patches might have less influence on the responses to objects than the responses to faces. Thus, we calculated face selectivity indexes under inactivation conditions and compared them with those under the control condition. The variance in the responses to faces and objects across runs, sessions, hemispheres, and monkeys was also taken into account in calculating the face selectivity index (see the Eq. 1). To limit the effects of the small sample size of sessions, we conducted a bootstrap resampling ($n = 10,000$) of the run set, which simulates distributions of standardized coefficients if the experiment were to be repeated with different runs or different treatments or different hemispheres or even different subjects. As shown in Fig. 5, inactivation of AF alone significantly modulated the face-selective index in MF ($p = 0.032$ uncorrected), with a similar trend in AL but not in ML, whereas inactivation of AL alone significantly modulated the face-selective index in AF ($p < 0.01$), with a similar trend in ML but not in MF. These findings indicated that the face selectivity in the middle face patches arises, in part, from top-down input from the anterior face patches and that there might exist a parallel organization among the face patches.

The impact of inactivation of the anterior face patches on ROIs in the hemispheres contralateral to the inactivation sites were also examined. Again, the above-mentioned findings in the ipsilateral hemispheres were not observed in the contralateral hemispheres (for group results, see Fig. 6; for individual hemisphere results, see Figs. S2B, 7–10B), suggesting that the inactivation mainly affected the hemispheres ipsilateral to the inactivation sites leaving the contralateral hemispheres unaffected.

Compared with data from the control sessions, combined AF and AL inactivations as well as AF inactivation alone, but not AL inactivation alone, significantly reduced responses to faces and objects in the ipsilateral amygdala ($p < 0.001$; Fig. 7A), whereas the responses in the contralateral amygdala were unaffected (Fig. 7B). These results indicate that the amygdala might mainly receive information from the face patch in the fundus of STS, AF, compared to the patch in the lateral portion of IT cortex, AL. This result further supports the functional organization along a fundal-lateral axis.

Again, no significant effects of inactivation of the anterior face patches were found on the object-selective regions (Figs. S11 and 12B).

### Functional relationship withinin and between the IT face network and amygdala

To explore further the functional relationships among the four IT face patches, we conducted regression analyses with responses to faces in one face patch as the outcome measure and responses to faces in the other face patches as predictors. As shown in Fig. 8A, in response to faces, the two face patches along the same fundal-lateral axis (e.g., MF and AF) predicted each other with the highest degree, while the two face patches along the posterior-anterior axis (e.g., MF and ML) predicted each other with the higher degree. However, face patches across the fundal-lateral and posterior-anterior axis (e.g., MF and AL, ML and AF) could not predict each other. The same analysis was conducted for responses to objects. Similar, albeit weaker, results were found to those obtained for responses to objects (see correspondence between faces and objects in Fig. 8B).

We also explored the functional relationships between the amygdala and the four face patches. As shown in Fig. 9, responses to faces in AF ($p = 0.001$) but not AL or MF or ML could predict responses to faces in the amygdala. Moreover, responses to objects in AF ($p = 0.001$) and, to a lesser degree (AF versus MF: $p = 0.045$ uncorrected; AF versus ML: $p = 0.047$ uncorrected), MF ($p = 0.033$) and ML ($p = 0.019$), but not AL, could predict responses to objects in the amygdala.

Together, these results indicate that, in the posterior-anterior extent, there were stronger relationships between fundal (AF and MF) and between lateral (AL and ML) face patches than across fundal-lateral face patches (i.e., between AF and ML, between AL and MF). Moreover, the responses in the amygdala depended on input from the face patch AF. The relationship within and between the IT face network and amygdala was largely mirrored when we examined the coefficients for predicting the neural response for objects in each region (Fig. 10).

## Discussion

In the present study, we measured neural responses to faces and objects using fMRI after transient pharmacological inactivation of specific face patches (MF, ML, AF, or AL, or pairs of these patches) in awake monkeys. Our results extend previous findings of the projections among face patches by identifying the functional roles of the face patches in the face-processing network. We revealed a bidirectional and parallel organization among the face patches within the IT cortex along the antero-posterior and fundal-lateral axes. Moreover, given the inactivation manipulations, we were able to make causal inferences regarding the direction of signal propagation within the IT face network as well as between the IT cortex and the amygdala by examining the impact of systematic inactivation of each patch on all the other

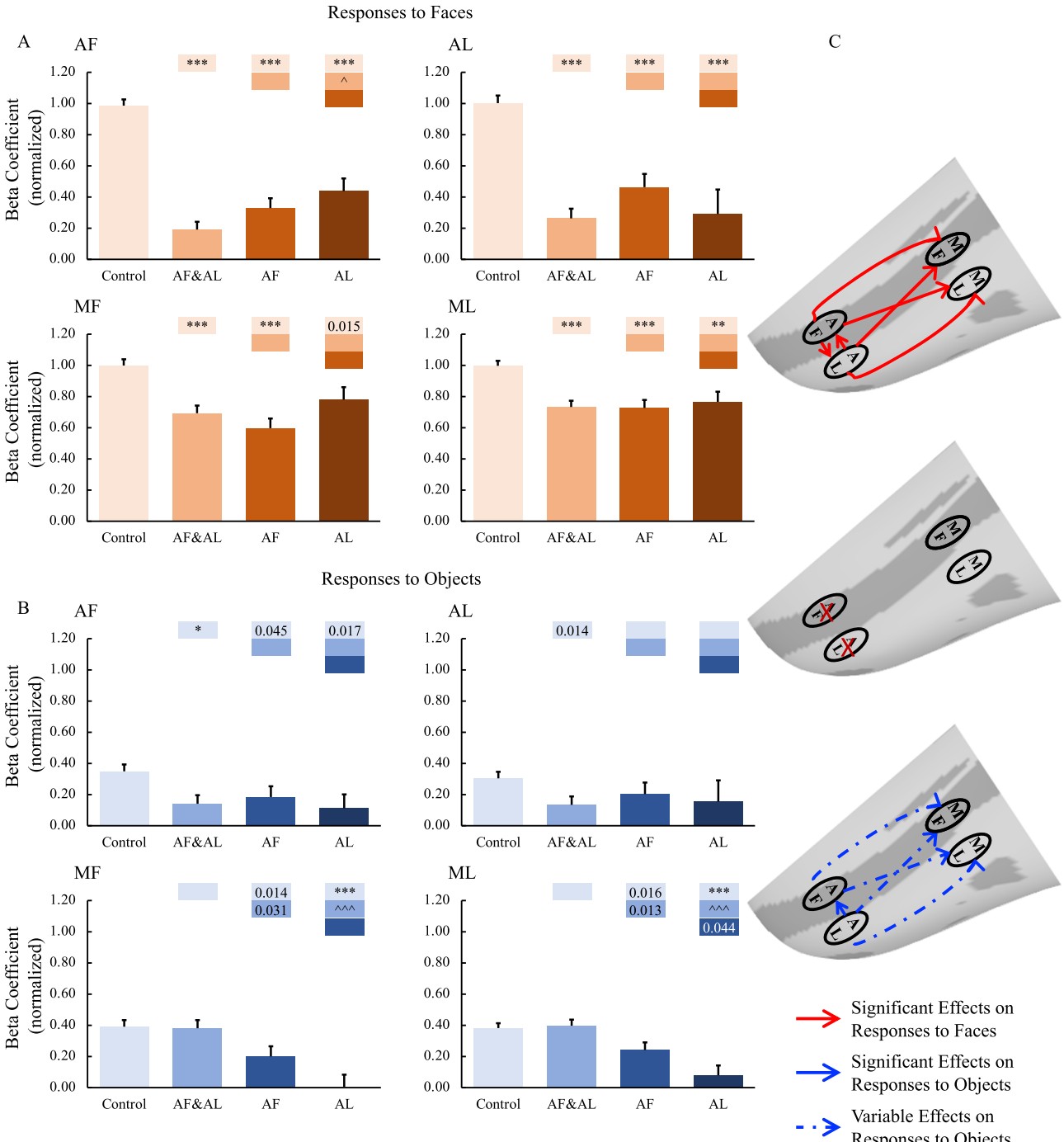

**Fig. 4 | Effects of anterior face patch inactivation on responses in the ipsilateral temporal face patches. A,B** Effects of anterior face patch inactivation on responses to faces and objects in the temporal face patches in the hemisphere ipsilateral to the inactivation sites across both monkeys. Box plots display mean values ± SEM. The name of the ROI is shown at the top left of each panel. AF&AL: combined AF and AL inactivation; AF: AF alone inactivation; AL: AL alone inactivation. Generalized Linear Mixed Models were performed for each ROI (AF, MF, and ML: $n = 486$; AL: $n = 336$) with Treatment and Hemisphere as fixed factors, and Monkey (C and D), L-R hemisphere (Left and Right), Session, and Run as random factors. Post hoc testing was done with correction for multiple testing using the Holm-Bonferroni method. Color bars above a particular histogram contain the statistical results of comparing this inactivation condition and the condition to its left with the same color as the bar: *, significant difference from control (*$p < 0.05$, **$p < 0.01$, ***$p < 0.001$, in light red/blue boxes); ^significant difference from combined AF and AL inactivation (^$p < 0.05$, ^^^$p < 0.001$, in middle red/blue boxes); value, uncorrected $p$ value; empty bar, no significant difference no matter whether multiple comparison corrections were done or not. **C** The schematic diagram of the effects of inactivation of the anterior face patches. The top and bottom panels show the significant or weak/variable (dashed lines) effects of middle face patch inactivation on responses to faces (red lines) and objects (blue lines) in the ipsilateral hemisphere, respectively. The middle diagram shows the location of the inactivation (marked with red X). MF middle fundal face patch, ML middle lateral face patch, AF anterior fundal face patch, AL anterior lateral face patch.

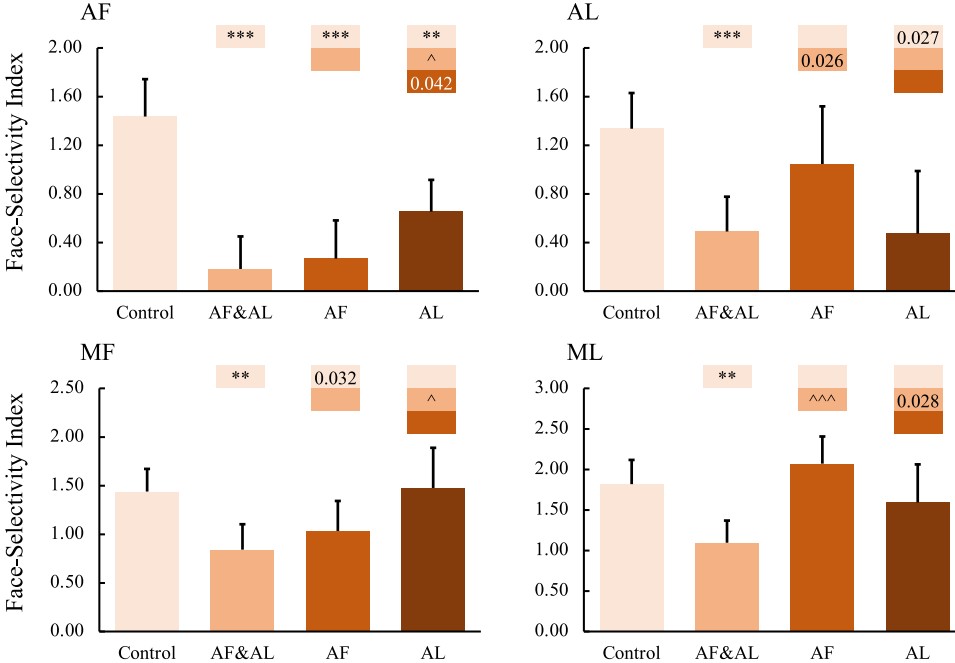

**Fig. 5 | Effects of anterior face patch inactivation on face-selectivity index in the ipsilateral temporal face patches.** The name of the ROI is shown at the top left of each panel. AF&AL: combined AF and AL inactivation; AF: AF alone inactivation; AL: AL alone inactivation. *P* values were calculated with bootstrap tests (*n* = 10,000) adjusted for multiple testing using the Holm-Bonferroni method. Color bars above a particular histogram contain the statistical results of comparing this inactivation condition and the condition to its left with the same color as the bar: *, significant difference from control (**$p < 0.01$, ***$p < 0.001$, in light red/blue boxes); ^significant difference from combined AF and AL inactivation (^$p < 0.05$, ^^^$p < 0.001$, in middle red/blue boxes); value, uncorrected *p* value; empty bar, no significant difference no matter whether multiple comparison corrections were done or not. Box plots display mean values ± 95% CI. MF middle fundal face patch, ML middle lateral face patch, AF anterior fundal face patch, AL anterior lateral face patch.

regions of interest. Below, we discuss the significance of these findings for understanding the topographic organization of face patches.

## A bidirectional organization among the IT face patches

Previous microstimulation and tracer studies have shown that the middle face patches send projections to the anterior face patches[6,7,28]. In the present study, we found that these projections played a necessary role in propagating visual input information to the anterior face patches: after unilateral inactivation of the middle face patches, responses to both faces and non-face objects in the anterior patches in the ipsilateral, but not contralateral, hemisphere were eliminated. These results suggest a sequence of specific processing stages, which is consistent with previous functional evidence of increasing abstraction of representation in more anterior temporal regions[9,29]. For example, it has been demonstrated that the intrinsic multivoxel response patterns to individual exemplars showed a better categorical distinction between faces and non-face objects in the anterior face patches than in the middle patches[29]. Our findings provide direct evidence for a serial hierarchy between the middle and anterior face patches.

Consistent with the fact that the pathways in the visual cortex are bidirectional[8], the middle face patches also receive projections from the anterior face patches. For example, stimulation of AL led to activation of ML[6,28], while stimulation of AF led to activation of MF[6]. Our results suggest a refinement of our understanding of feedback within the IT face network: inactivation of the anterior face patches mainly affected the responses to faces but, only minimally, if at all, the responses to objects in the middle face patches. That is, the face selectivity in the middle face patches arises, in part, from top-down input from the anterior face patches (Fig. 5). This is consistent with electrophysiological recordings within MF and ML as, in these two areas, responses to contrast-inverted faces were slower than those to control faces, possibly due to the involvement of feedback to resolve

the contrast inversion[30]. Our results reveal the function of these feedback projections and provide empirical evidence of their importance in face processing.

Taken together, our findings demonstrate clear differences in the functional roles of projections between the middle face patches and the anterior face patches, indicating that the face-processing network is organized along a posterior-anterior axis with a bidirectional dialogue.

## A parallel organization among the IT face patches

Although multiple face patches have been targeted in previous microstimulation and tracer studies to investigate the circuits underlying face processing, the two patches, MF and AF, within the STS fundus, have rarely been accessed. One previous study did find that stimulation in AF elicited activation in MF in one monkey[6], however, such findings may result from direct or indirect connections. Here, we investigated the effects of MF and AF inactivation, respectively, to explore the functional organization along the fundal-lateral axis within the IT face network. Our findings showed that inactivation of MF alone substantially reduced (or even eliminated) responses in AF, while inactivation of AF alone reduced the face selectivity in MF. Moreover, under the different types of middle face patch inactivations (i.e., combined MF and ML inactivation, MF alone inactivation, and ML alone inactivation), the correlation analysis demonstrated that response patterns of MF were similar to those of AF. Finally, our results showed that responses in MF could predict responses in AF, and vice versa. Therefore, our findings provide multiple strands of evidence for the existence of connections between MF and AF, and, hence, provide the missing link regarding the pattern of connections among the face patches. We also found a privileged dialogue between face patches of the lateral portions of IT cortex as reported in previous anatomical studies[6,7]: inactivation of ML or AL eliminated/reduced responses in each

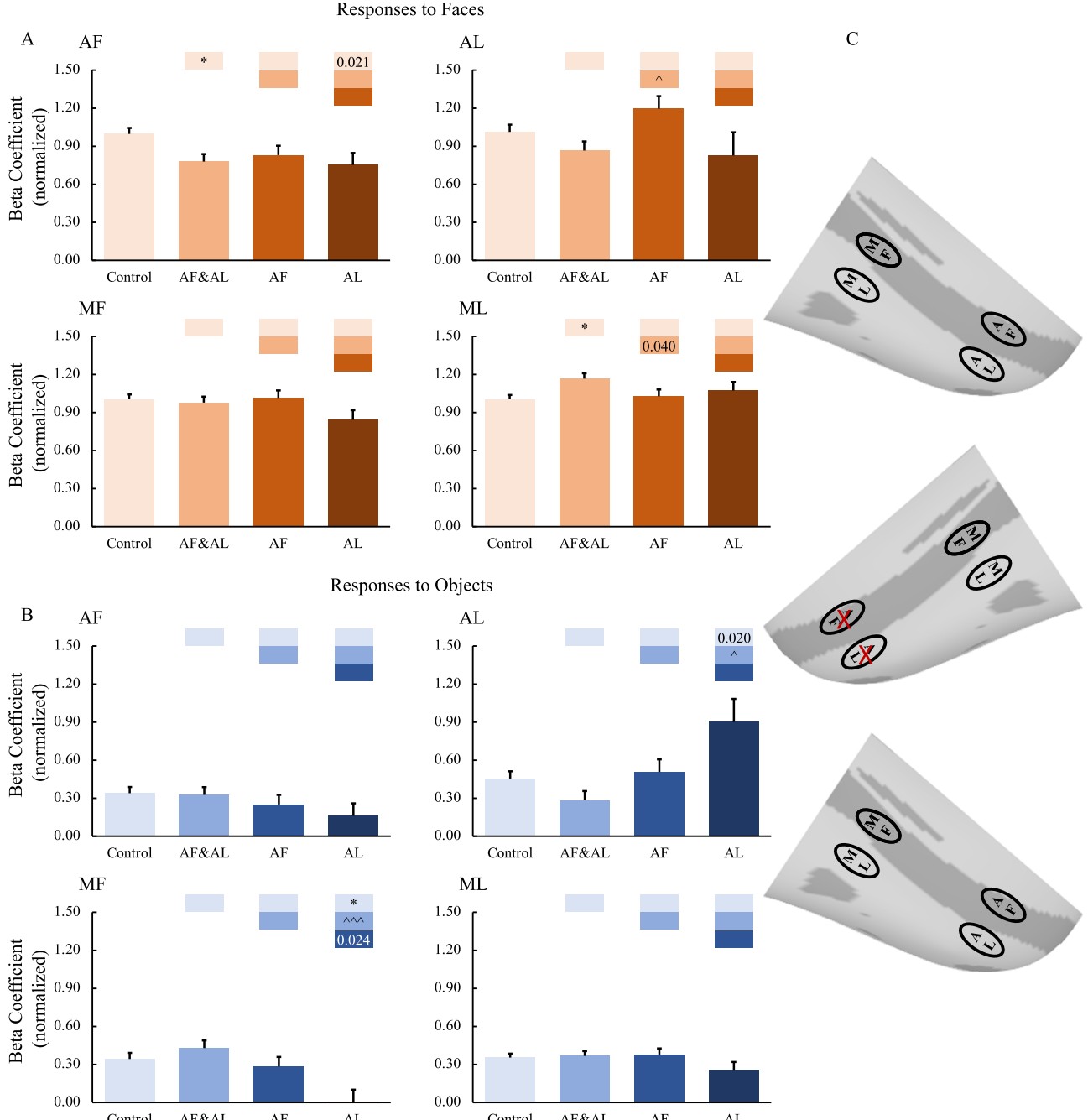

**Fig. 6 | Effects of anterior face patch inactivation on responses in the contralateral temporal face patches. A**, **B** Effects of anterior face patch inactivation on responses to faces and objects in the temporal face patches in the hemisphere contralateral to the inactivation sites across both monkeys. Box plots display mean values ± SEM. The name of the ROI is shown at the top left of each panel. AF&AL: combined AF and AL inactivation; AF: AF alone inactivation; AL: AL alone inactivation. Generalized Linear Mixed Models were performed for each ROI (AF, MF, and ML: n = 486; AL: n = 336) with Treatment and Hemisphere as fixed factors, and Monkey (C and D), L-R hemisphere (Left and Right), Session, and Run as random factors. Post hoc testing was done with correction for multiple testing using the Holm-Bonferroni method. Color bars above a particular histogram contain the statistical results of comparing this inactivation condition and the condition to its left with the same color as the bar: *, significant difference from control (*p < 0.05, in light red/blue boxes); ^ significant difference from combined AF and AL inactivation (^p < 0.05, ^^^p < 0.001, in middle red/blue boxes); value, uncorrected p value; empty bar, no significant difference no matter whether multiple comparison corrections were done or not. **C** The schematic diagram of the effects of inactivation of the anterior face patches. The top and bottom panels show no significant effects of middle face patch inactivation on responses to faces and objects in the contralateral hemisphere, respectively. The middle diagram shows the location of the inactivation (marked with red X). MF middle fundal face patch, ML middle lateral face patch, AF anterior fundal face patch, AL anterior lateral face patch.

other; more importantly, responses of ML and AL could be well predicted by each other.

There are several possible scenarios that could account for the above-mentioned parallel connections (between MF and AF as well as between ML and AL): (1) indirect parallel connections through regions other than face patches, (2) indirect parallel connections through

other face patches, or (3) direct parallel projections. Previous microstimulation and tracer studies have indicated that the first scenario is unlikely to hold[6,7]. Although inactivation of each of the four targeted face patches influenced all the other three face patches, the parallel connections between lateral areas (ML and AL) and between fundal areas (MF and AF) were more pronounced than those that crossed

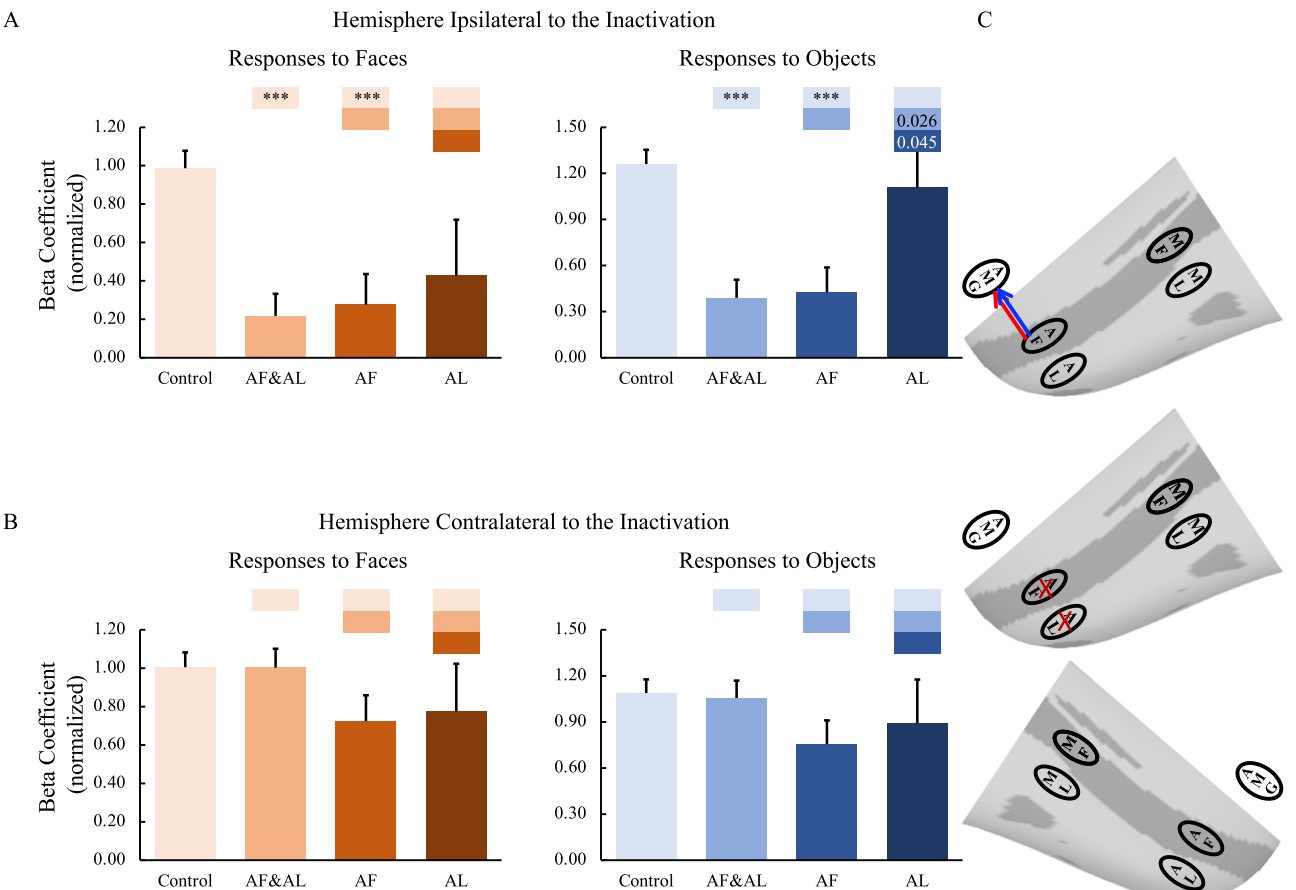

**Fig. 7 | Effects of anterior face patch inactivation on responses in the amygdala.** **A**, **B** Effects of anterior face patch inactivation on responses to faces and objects in the amygdala in the hemispheres ipsilateral and contralateral to the inactivation sites across both monkeys. Box plots display mean values ± SEM. AF&AL: combined AF and AL inactivation; AF: AF alone inactivation; AL: AL alone inactivation. Generalized Linear Mixed Models were performed for each ROI (AF, MF, and ML: $n = 486$; AL: $n = 336$) with Treatment and Hemisphere as fixed factors, and Monkey (C and D), L-R hemisphere (Left and Right), Session, and Run as random factors. Post hoc testing was done with correction for multiple testing using the Holm-Bonferroni method. Color bars above a particular histogram contain the statistical results of comparing this inactivation condition and the condition to its left with the same color as the bar: *, significant difference from control (***$p < 0.001$, in light red/blue boxes); value, uncorrected $p$ value; empty bar, no significant difference no matter whether multiple comparison corrections were done or not. **C** The schematic diagram of the effects of inactivation of the anterior face patches. The top and bottom panels show the effects of anterior face patch inactivation on responses to faces and objects in the hemispheres ipsilateral (red line: significant effect on responses to faces; blue line: significant effect on responses to objects) and contralateral to the inactivation sites, respectively. The middle diagram shows the location of inactivation (marked with red X). MF middle fundal face patch, ML middle lateral face patch, AF anterior fundal face patch, AL anterior lateral face patch, AMG amygdala.

between lateral to fundal regions (between MF and AL as well as between ML and AF). Thus, our findings indicate that, between the face patches, the cross connections, if they exist, are functionally weaker and less influential than the parallel connections. Such a parallel organized face-processing network has been implied by previous studies but no direct empirical evidence was available. For example, microstimulation studies have found that ML is connected to AF but more variably than to AL: stimulation in ML elicited activation in AF only in one monkey but in AL in all tested monkeys (three monkeys in[6], two monkeys in[28]); and stimulation in AF showed weaker activation in ML than to MF[6]. Tracer results found projections from AF to ML but only in one of three injected monkeys[7]. Similarly, the connections between MF and AL were also weak or variable. Although stimulation of AL elicited activation in MF in one study[6], another study did not reveal consistent results[28]. Furthermore, injection of retrograde tracers in AL showed projections from MF that were weaker than to ML (e.g., 642/98 cells in MF versus 3025/218 cells in ML)[7].

In the present study, we found that inactivation of one middle face patch affected the responses (especially to faces) in another middle face patch, and the same held for the anterior face patches. Previous microstimulation studies observed that stimulation in ML elicited activation in MF and stimulation in AL elicited activation in AF[6,28]. Moreover, the tracer study also showed the projections from MF to ML as well as from AF to AL[7]. As we noted, these connections are not as robust as those between MF and AF and between ML and AL. Together with the above results, it appears that there may be some, albeit limited, communication between the fundal and lateral pathways. Human neuroimaging findings have shown that the lateral pathway may propagate static (form) information to the fundal pathway to support dynamic information processing[14].

Taken together, our findings demonstrate that the face-processing network is also organized in parallel along a fundal-lateral axis. These parallel processing pipelines may provide the anatomical basis accounting for previous functional findings. For example, the effects of electrical microstimulation on the perception of face identity in AF and MF were similar but smaller than those elicited by stimulating ML and AL[31]. The fundus face patches (MF and AF) were more sensitive to facial expression, whereas the lateral face patches (ML and AL) were more sensitive to head orientation[32]. Moreover, it has also been shown that the fundus regions of the STS but not the lateral face patches are linked to facial motion[15,16]. Neural models of face processing based on fMRI studies in humans have suggested that there are two neural

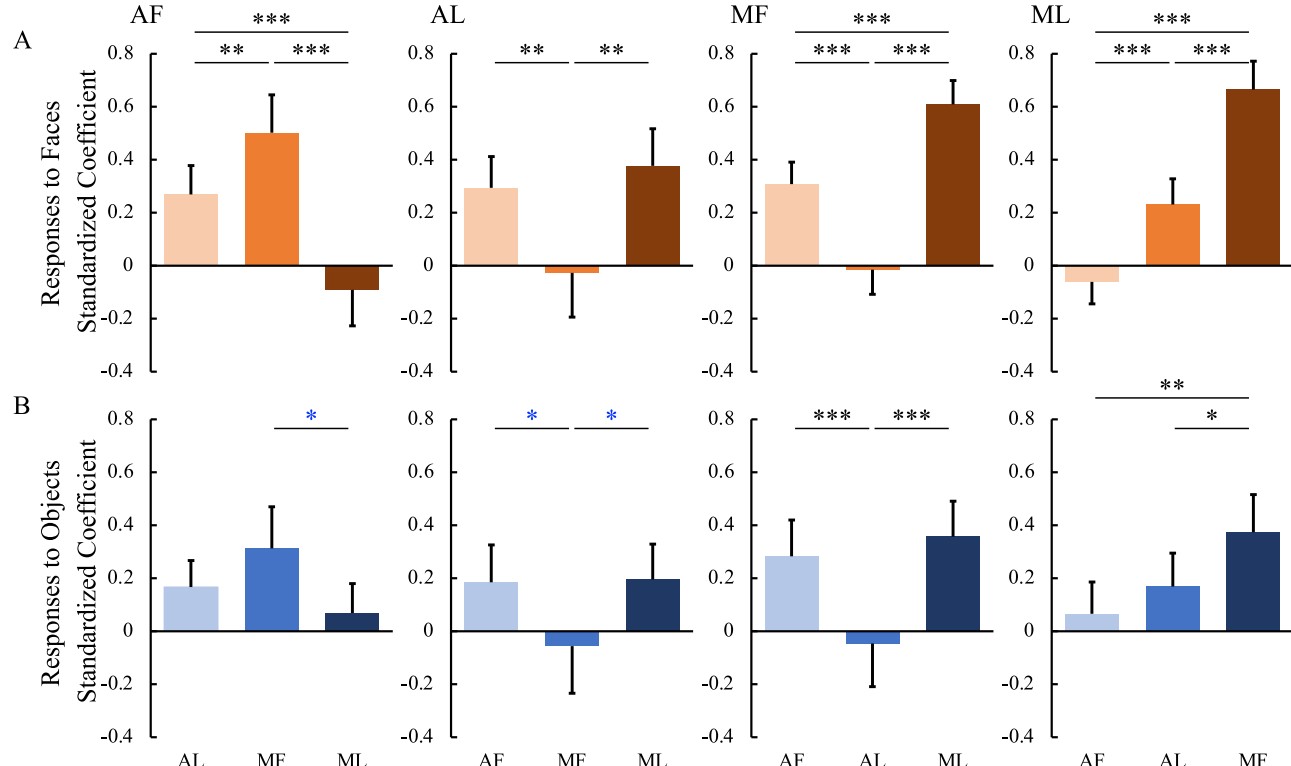

**Fig. 8 | Functional relationships among the temporal face patches.** Standardized beta-coefficients for linear regression with responses to faces (**A**) and objects (**B**) in one face patch (the name is shown at the top left of each panel) as outcome and responses to faces/objects in the rest of the face patches as predictors (the name is shown as the horizontal axis labels at the bottom of each column). *P* values were calculated with bootstrap tests ($n = 10,000$) adjusted for multiple testing using the Holm-Bonferroni method. Black *$p < 0.05$, **$p < 0.01$, ***$p < 0.001$. Blue *$p < 0.05$ uncorrected. Box plots display mean values ± 95% CI. MF middle fundal face patch, ML middle lateral face patch, AF anterior fundal face patch, AL anterior lateral face patch.

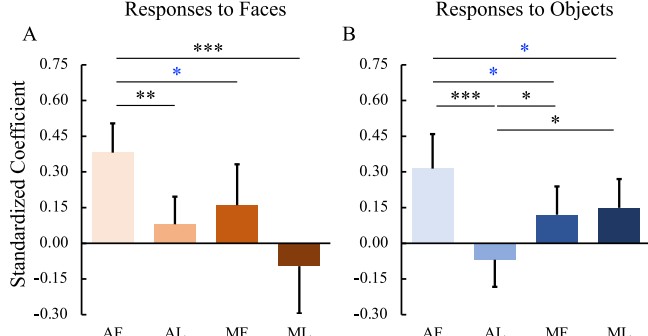

**Fig. 9 | Functional relationships between the amygdala and the temporal face patches.** Standardized beta-coefficients for linear regression with responses to faces (**A**) and objects (**B**) in the amygdala as outcome and responses to faces/objects in each of the temporal face patches as predictors (the name is shown as the horizontal axis labels at the bottom of each column). *P* values were calculated with bootstrap tests ($n = 10,000$) adjusted for multiple testing using the Holm-Bonferroni method. Black *$p < 0.05$, **$p < 0.01$, ***$p < 0.001$. Blue *$p < 0.05$ uncorrected. Box plots display mean values ± 95% CI. MF middle fundal face patch, ML middle lateral face patch, AF anterior fundal face patch, AL anterior lateral face patch.

pathways of face processing: the dorsal pathway from the occipital face area to the superior temporal sulcus is mainly involved in the processing of dynamic information in faces, while the ventral pathway from the occipital face area to the fusiform gyrus processes the form information in faces[13,14,25,33]. The possibility of correspondences between macaque and human face-selective regions along the dorso-ventral axis (fundal-lateral axis in monkeys) has been proposed based on functional similarities[34]. Our results provide empirical evidence for

such parallel fundal-lateral organization in the monkey face processing system and provide further evidence for the homologies between macaque and human face-processing systems, involving multiple functional compartments.

Beyond MF, ML, AF, and AL, several other face patches, which include posterior lateral (PL) and anterior medial (AM) face patches[3], as well as two recently discovered patches for familiar face processing[35,36], have also been identified in the temporal lobe. It might thus be interesting in future studies to investigate the relationship of these regions with the regions of interest covered in our studies (MF, ML, AF, and AL) to understand the mechanisms at play during face perception.

### Connections between the amygdala and face patches
In the present study, we also investigated the interplay between the IT face patches and the amygdala, a key structure in emotional processing that receives inputs from the IT cortex[19,20,26]. Here, the goal was to determine whether the fundal-lateral organization in the IT face network extends to the amygdala. Our results indeed revealed that inactivations of face patches, especially AF, reduced amygdala responses to faces and objects and that responses in AF, but not AL, could predict responses to both faces and objects in the amygdala, indicating that the responses in the amygdala depended on input from the face patch AF. This is consistent with previous functional evidence showing that the fundus face patches communicated with the amygdala to support the processing of socially-related information (e.g., facial expressions and social interaction)[4,37].

### Interplay between the IT face patches across hemispheres
In the present study, the impact of face patches inactivation was mostly restricted to the ipsilateral hemisphere, with minimal impact on

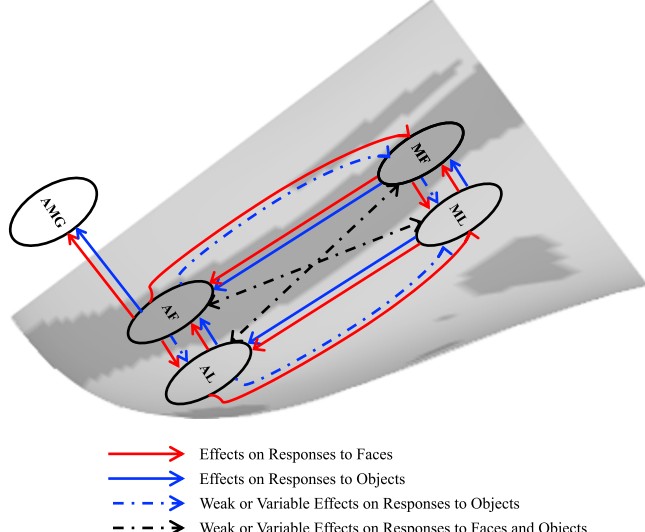

Effects on Responses to Faces
Effects on Responses to Objects
Weak or Variable Effects on Responses to Objects
Weak or Variable Effects on Responses to Faces and Objects

**Fig. 10 | Summary of the internal connections of the face patches.** The direction of the arrows indicates the direction of effects of inactivation. Red lines, effects on responses to faces; Blue lines, effects on responses to objects; Blue dashed lines, weak or variable effects on responses to objects; Black dashed lines, weak or variable effects on responses to both faces and objects. MF middle fundal face patch, ML middle lateral face patch, AF anterior fundal face patch, AL anterior lateral face patch, AMG amygdala.

the contralateral hemisphere of the injection sites. This is compatible with previous studies. In particular, anatomical studies found weaker connections with face patches in the contralateral hemisphere of the injection sites compared to the ipsilateral hemisphere[7]. Microstimulation of face patches evoked weaker and inconsistent activation in face patches contralateral to the stimulation sites[6]. While these data indicate that interhemispheric transcallosal connectivity exists, its functional significance remains to be determined. A recent study found that inactivation of the middle face patch induces behavioral deficits in face recognition only in the contralateral (but not in the ipsilateral) hemifield[38]. In the present study, since we did not require animals to perform any complex face-related tasks, we cannot know whether the face network in one hemisphere suffices for processing face information. Further studies combining face patches inactivation and well-designed behavioral tasks will help shed light on the interplay between face patches across hemisphere.

### Extent of inactivation

The spread of muscimol is an important factor that might have affected the precision of the findings in the present study. Prior to this study, we had carefully tested the extent of muscimol spread in order to select a reasonable dosage. The current dosages were chosen based on these preliminary explorations (see[39]) to yield effective inactivation of face patches but with little inactivation of surrounding regions. On average, we found that of the total injection volume, 81.74% on average (MF: 88.93%; ML: 85.87%; AF: 79.25%; AL: 73.54%) was contained within the targeted face-selective ROIs, a value comparable to previous studies (e.g., 76% in[40]), Furthermore, the differences in results following inactivation of patches in the fundus versus lateral portions of the STS, which were close to each other, also indicated that the injected muscimol was mainly limited to the targeted ROIs (also see Figs. S14–17). For example, the results of MF inactivation were different from those of ML inactivation (Fig. 2A), indicating that the injected muscimol rarely leaked or 'bled over' to neighboring face patches. Taken together, in the present study, the injected muscimol primarily silenced the targeted face patches but left the surrounding regions less (if at all)

affected. That is, the potential muscimol leakage is likely minimal and should not affect our major results.

We combined systematic and selective inactivation of the IT face patches with fMRI in monkeys and demonstrated a bidirectional, hierarchical organization of face patches in the macaque IT cortex (Fig. 10): the anterior face patches require inputs from the middle face patches for their responses to both faces and objects, while the face selectivity in the middle face patches arises, in part, from top-down inputs from the anterior face patches. Our findings confirm the functional bidirectional interplay between the middle and anterior face patches. No influence of inactivation of the middle and anterior face patches was observed in object-selective regions, attesting to the specificity of the results. Importantly, we also outline a parallel functional organization along a fundal-lateral (or dorso-ventral) axis in the IT face patches system. Interestingly, beyond the IT cortex, we found that information from the fundal pathway (especially from AF) is tightly linked to the amygdala. These findings extend our fundamental understanding of the dynamics that govern the neural circuits underlying face processing.

## Methods

### Subjects and general procedures

Two male macaque monkeys participated in these experiments (Monkeys C and D, *Macaca mulatta*, 9 years old; 6.5–7.5 kg). All procedures followed the Institute of Laboratory Animal Research (part of the National Research Council of the National Academy of Sciences) guidelines and were approved by the NIMH Animal Care and Use Committee. The monkeys were acquired from the same primate breeding facility in the United States, where they had social group histories as well as group-housing experience until their transfer to the National Institute of Mental Health (NIMH) for quarantine at ~4 years old. After that, they were individually housed with auditory and visual contact with other conspecifics in the same colony room. Each monkey was surgically implanted with a magnetic resonance (MR)-compatible head post under sterile conditions, using isoflurane anesthesia. After recovery, the monkeys were trained to sit in a plastic chair and fixate a central target for long durations with stable head position, facing a screen on which visual stimuli were presented[29,41].

### Brain activity measurements

Functional and anatomical MRI scanning was carried out in the Neurophysiology Imaging Facility Core [NIMH, National Institute of Neurological Disorders and Stroke (NINDS), National Eye Institute (NEI)]. Before each scanning session, a contrast agent [monocrystalline iron oxide nanocolloid (MION)] was injected into the femoral or external saphenous vein (12–15 mg/kg) to increase the contrast/noise ratio and to optimize the localization of fMRI signals[42]. Imaging data were collected in a 4.7 T Bruker scanner with a surface coil array (two elements in Monkey C and eight elements in Monkey D). Twenty-eight 1.5 mm coronal slices (no gap) were acquired using single-shot interleaved gradient-recalled echo planar imaging. Imaging parameters were as follows: voxel size: 1.5 mm isotropic; field of view: 96 × 54 mm; matrix size: 64 × 36; echo time (TE): 13.8 ms; repetition time (TR): 2 s; flip angle: 90°. A low-resolution anatomical scan was also acquired in each session to serve as an anatomical reference and demarcate the spread of muscimol [modified driven equilibrium Fourier transform (MDEFT)] sequence, voxel size: 1.5 × 0.5 × 0.5 mm; field of view: 96 × 96 mm; matrix size: 192 × 192; TE: 3.95 ms; TR: 11.25 ms; flip angle: 12°). To facilitate cortical surface alignment and the following local targeting of regions, we also acquired high-resolution T1-weighted whole-brain anatomical scans in separate sessions, using the MDEFT sequence. Imaging parameters were as follows: voxel size: 0.5 mm isotropic; TE: 4.1 ms; TR: 12 ms; flip angle: 12°.

## Experimental design and task

To identify Regions of Interest (ROIs), we performed an initial localizer experiment[29,41]. The stimuli were presented in a block design. For Monkey C, grayscale photos of neutral monkey faces, familiar places, familiar objects, Fourier-phase scrambled faces, Fourier-phase scrambled places, and Fourier-phase scrambled objects were presented in separate blocks (see Fig. 1A for example stimuli). Each block lasted 30 s and was presented once in each run. For Monkey D, grayscale photos of neutral monkey faces, familiar places, familiar objects, and Fourier-phase scrambled faces were presented in separate blocks. Each block lasted 32 s and was presented twice in each run.

In the subsequent inactivation and corresponding control experiments, to optimize the statistical power, only Grayscale photos of neutral monkey faces and familiar objects were presented to the animals. Each categorical block lasted 32 s and was presented four times in each run. In all experiments, each categorical block alternated with 20 s fixation blocks. Individual runs began and ended with a fixation block. Different pseudorandom sequences were used in each run. In each categorical block, 16 images were each presented for 700 ms followed by a 300 ms interval and repeated twice. All stimuli used in this experiment were identical to those used in[29]. Stimuli spanned a visual angle of 11° (maximal horizontal and/or vertical extent) on a uniform gray background and were presented foveally with a red fixation square (0.2°) superimposed on each image. Eye position was monitored with an infrared pupil tracking system (iView, Inc). The monkeys were required to maintain fixation on the red square superimposed on the stimuli to receive a liquid reward. In the reward schedule, the frequency of reward increased as the duration of fixation increased[29,41]. Rewards were controlled by a QNX system. Data were included from only those runs in which fixation was maintained on at least 90% of the runs. The stimuli were presented using the Presentation software (version 12.2, www.neurobs.com).

## fMRI data preprocessing

Functional data were preprocessed using Analysis of Functional NeuroImages software (AFNI 20.2.10)[43]. Images were realigned to the base volume of one initial localizer session. Then, the data were smoothed with a 2 mm full-width half-maximum Gaussian kernel. Signal intensity was normalized to the mean signal value within each run. For each voxel, we performed a single univariate linear model fit to estimate the response amplitude for each condition. The model included a hemodynamic response predictor for each category and regressors of no interest (baseline, movement parameters from realignment corrections, and signal drifts). A general linear model and a MION kernel were used to model the hemodynamic response function[42]. All fMRI signals throughout the paper have been inverted so that an increase in signal intensity indicates an increase in activation. Each monkey's statistical results were projected onto its own inflated cortical surfaces.

## Definition of face-selective ROIs, the amygdala and object-selective regions

A two-step ROI definition method was conducted in the present study. First, to define ROIs, all runs were concatenated across the initial localizer. Each monkey was scanned in 2–3 localizer sessions, resulting in a total of 44 runs (44 category repetitions) for Monkey C and 23 runs (46 category repetitions) for Monkey D. For each monkey, we identified face patches using the contrast of neutral monkey faces versus familiar objects ($p < 0.005$ uncorrected; FDR corrected $q$ value is 0.0056 for monkey C and 0.0209 for monkey D). The same threshold was applied to localize the object-selective regions using the contrast of familiar objects versus neutral monkey faces. Consistent with previous studies[2,29,44], this contrast yielded a set of face patches in IT cortex in each hemisphere, in each subject. Note that it was not always possible to separate precisely the 6 face patches in an individual monkey either because of poor signal in the anterior temporal lobe

(especially the ventral surface) or differences in patterns of patches across monkeys. For example, we did localize the AM patch in Monkey C (Fig. S18) but not in Monkey D (Fig. S19). AM is on the ventral surface of IT, which is the first and most affected area by the MION accumulation. Therefore, in Monkey C, presumably because of this accumulation, we were unable to localize AM in many later sessions. Moreover, since we could not localize AM in Monkey D (which may be caused by relatively weaker signal in this area and individual differences) even in the initial localizer experiment, we did not include AM in the present study. We conservatively focused on 4 patches as inactivation targets. Specifically, we chose two middle face patches near area TEO, one located in the fundus of the STS ("MF," for middle fundus) and one on the lower lip of the STS ("ML," for middle lateral); and two more anterior patches in area TE, one located near the fundus of the STS ("AF", for anterior fundus), and one on the lower lip of the STS ("AL", for anterior lateral). Within each face patch, the peak selective voxel was initially identified and targeted for the infusion. Moreover, all face-selective voxels within a radius of 3 mm around the peak voxel were combined to define the face-selective ROIs. In addition, we identified the peak of activation in the amygdala using the face-responsive maps ($p < 0.001$ uncorrected): all face-responsive voxels within a radius of 3 mm around the peak voxel and anatomical ROI of the amygdala were combined to yield the amygdala ROIs.

Due to MION accumulation, not all the voxels in these defined ROIs survived for inclusion in the subsequent inactivation experiments. If we included these unresponsive voxels in our final analyses, the mean signals would likely be weak even in the control conditions, and then comparisons between the control and inactivation conditions would be hampered. For example, when we conducted inactivations in the left hemisphere in Monkey D, no voxels were obtained at the selected threshold ($p < 0.005$ uncorrected) from the initially-defined LAL and the mean beta coeffects of the initially-defined LAL was close to 0 even in the control conditions. Therefore, we conducted a second step: we concatenated all the control sessions under the same set of inactivations (Table S5, M-Controls and A-Controls in each column), and then identified the face-selective voxels ($p < 0.001$ uncorrected). Any voxels within the initially-defined ROIs that could not be identified in the corresponding control sessions were removed to yield the final ROIs for each type of inactivation sessions. Though this two-step ROI definition method may not be ideal, the approach to defining ROI should not affect the results substantially.

## Local targeting

A grid system was used to achieve precise targeting of the intracerebral injections, as described in detail elsewhere[39]. Briefly, the monkeys were anesthetized and surgically implanted with a customized rectangular chamber (52 mm × 32 mm) made of Ultem, using aseptic techniques. After at least 2 weeks of recovery, the chamber was filled with a dilute gadolinium solution [Magnevist, Berlex Pharmaceuticals; 1:1200 v:v in sterile saline (1 part Magnevist to 1200 parts saline), pH 7.0–7.5], and the appropriate guide grid was inserted into the chamber to visualize the grid holes. High-resolution T1-weighted whole-brain anatomical scans (see *Brain Activity Measurements*) were obtained to map the grid. Target coordinates were confirmed in a subsequent scan session with local infusions of sterile dilute gadolinium (Gd) contrast agent (2.7–3.75 μL of a 5 mM solution[45]); this contrast agent was also included to visualize the injection spread in each of the functional injection scanning sessions (Figs. S14–17), as the cortical spread of Gd closely tracks the diffusion extent of muscimol[46,47]. Gd mainly affects the T1-weighted signals and not T2-weighted signals and, thus, does not significantly influence the functional signals[46,47]. To control for any potential effects of Gd on the fMRI activity (though it may be very weak), in the control conditions, we injected the same volume of Gd into the target region as was used for the Gd plus

muscimol condition; thus, the differences between control and inactivation sessions should mainly have been caused by muscimol.

## Transient inactivation

Reversible inactivations were achieved by infusing muscimol (18 mM, 2.7–3.75 μL, sterile filtered) into each target in the awake animals[48]. The dosages, which were comparable to (and even lower than) the dosages used in previous studies in monkeys[40,48], were chosen to yield effective inactivation of face patches but with little inactivation of surrounding regions. We made microinfusions at the rate of 0.18 μl/min (e.g., for the dosage of 2.7 μl, the injection takes 15 min to complete), which was similar to or even slower than that used in previous studies (e.g., 0.2 μl/min in[49], average rate of 0.4 μL/min in[40]). After the injection, we waited for 12–15 min before retracting the cannula to avoid any spread of muscimol along the track of the cannula.

The injections were performed outside the scanner with metal cannulas. When the injection was done (15–21 mins depending on the injected volume), we waited 15–20 mins (also depending on the injected volume) before removing the cannula. Then, we removed the injection grid, cleaned the chamber, transferred animals to the scanner room, and started the setup for the scanning. Therefore, the scanning started about 50 min after the injection. To get a clear image of the Gd injection spread, we collected a T1w image first. The fMRI data collection lasted about 1.25 h (~10 runs). Thus, the fMRI data were acquired between ~1 and 2.5 h from the end of the injection procedure, falling into the window of the maximal muscimol effects (~0.5 h to ~4 h)[50–52].

By co-infusing the MR contrast agent Gd, the extent of muscimol spread is detectable on the anatomical MR scan (T1w) and the Gd-signal was used to demarcate the spread of muscimol. We analyzed the overlap between the inactivation injections and ROIs as described in a previous study[40]. Briefly, ROIs based on the initial localizer experiments drawn on EPI images were resampled to the resolution of T1w images. Then, we calculated the volume (in voxels) of the injections as well as ROIs, and the overlap between the injections and ROIs. The volume of injections was defined as all the voxels with values greater than 50% of the maximum value of Gd brightness on T1w images within a radius of 2.5 mm around the injection site. The injection overlap was defined as the volume of the overlap divided by the volume of the injection, while the ROI overlap was defined as the volume of overlap divided by the volume of the targeted ROI. Theoretical ROI overlap was defined as the volume of the injection divided by the volume of the targeted ROI assuming concentric spheres for both volumes (Tables S2 and 3).

The microinfusions targeted one (AF/AL/MF/ML) or two face patches (both AF and AL or both MF and ML) in one hemisphere per session. The order of inactivation sites was randomized across monkeys and hemispheres. The results from the muscimol-induced inactivation sessions were compared with those obtained from vehicle infusion sessions (Gd + saline). For a given period of time, we performed inactivation of anterior or middle face patches in one hemisphere. For example, inactivations of either AF or AL or both AF and AL in the right hemisphere were conducted over one period. This was done so that we could use the same set of control sessions for all three types of inactivations and then compare data across inactivation sites. For details on scan information for the different types of face patch inactivations, see Table S5. Note that the numbers of sessions per injection site were not equal. We applied the following two strategies in the experiment: (1) to keep the inactivation sessions and control sessions interleaved (1 control then 2–3 inactivations or so) to limit the effects of periods on MION accumulation, and so on; (2) to do the same set of inactivation (e.g., M which including combined MF and ML inactivation, MF alone inactivation, ML alone inactivation) together to perform the comparisons among them and between them within the same control session set. Only those instances, in which we confirmed

that the injection was successful (reaching the target sites well) and the spread of Gd was ideal (e.g., covering most of the ROIs), were used in the final analyses. We considered the animal's health, the problem of MION accumulation, and the possibility of reaching the desired target successfully each time and, after several failed attempts, we decided to move on to another injection site. In this case, we did get more control sessions due to the above strategy #1 and we used them all to increase the statistical power. The differences in the number of sessions and runs did cause some variability in the observed individual (hemisphere) patterns. Therefore, all the data from both monkeys and hemispheres were considered together to increase statistical power and reliability.

## Responses to faces and objects in the inactivation and corresponding control sessions

The signal was extracted from face-selective ROIs. We then calculated the response to faces and objects within each run (averaged across four repetitions). Note that due to considerations mentioned above (e.g., the possibility of reaching the desired target successfully), not all types of inactivations (for each type of anterior and middle face patch inactivations: F, L both F and L) were performed in all four hemispheres (see Table S5 for number of sessions for each type of inactivation and further details). To account for inter-session differences due to factors such as MRI coil placement and contrast agent clearance, first, the fMRI signals were normalized by separately scaling each run, which was achieved by dividing the signal change of each ROI by the maximal signal change measured in the visual cortex (in STS or V4)[15]. In the present study, there were no obvious hemispheric asymmetries in either the initial localizer experiments or in the inactivation experiments. Therefore, for the group analyses, the results from hemispheres ipsilateral and contralateral to the inactivation sites were collapsed respectively in the following analyses. To compensate for hemisphere and individual differences, when combining results from different hemispheres/subjects, the signal changes were normalized by separately scaling each hemisphere/subject, by dividing the average signal changes of corresponding control sessions evoked by faces. Because of the small sample size, a typical fixed effects model for monkey fMRI studies was used.

To explore the differences in treatment and hemisphere, we performed Generalized Linear Mixed Models (GLMMs) on the data from inactivations of middle face patches and anterior face patches separately using SPSS (v24) software (SPSS Inc., Chicago, Illinois, USA). Treatment (Control, F inactivation, L inactivation, and combined F and L inactivation) and Hemisphere (ipsilateral and contralateral hemisphere to the injection sites) were treated as fixed factors, and Monkey (C and D), L-R hemisphere (Left and right), Session, and Run were random factors. We then followed up with post hoc tests on responses to faces/object in each ROI, with adjustment for multiple testing using the Holm-Bonferroni method. We note here that all $p$ values are corrected unless specified otherwise.

## Face selectivity index

For each ROI, we calculated the selectivity index for its preferred category (i.e., faces) following the equation where μ and σ are the mean and standard deviation of the response (Eq. 1)[53]. The source of the variance in the preferred and nonpreferred distributions come from pooling beta coefficients across runs and sessions for a given ROI. The effect of treatments (4 levels: Control, F inactivation, L inactivation, and both F and L inactivation) on the selectivity index in each ROI was calculated with bootstrap tests adjusted for multiple testing using the Holm-Bonferroni method.

## Relationship among face patches and the amygdala

To assess further the causal relationship among the face-selective ROIs and amygdala, we conducted linear regression analyses. We modeled

the relationship between one ROI and other ROIs by fitting them into a linear equation (Eq. 2). For example, the relationship between AF with AL, MF, and ML when subjects were viewing faces was obtained by

$$Y = a + b_1 X_1 + b_2 X_2 + b_3 X_3 + \varepsilon \qquad (2)$$

Where Y is the responses to faces in AF, $X_1$ is the responses to faces in AL, $X_2$ is the responses to faces in MF, and $X_3$ is the responses to faces in ML. All the runs across all the treatments (control, combined AF and AL inactivation, AF alone inactivation, AL alone inactivation, combined MF and ML inactivation, MF alone inactivation, ML alone inactivation) and across two hemispheres from both monkeys were used in the model ($n = 575$, see Table S5). To assess differences between standardized coefficients ($b_1, b_2, b_3$ in the above equation), we conducted a bootstrap resampling ($n = 10,000$) of the run set, which simulates distributions of standardized coefficients if the experiment were to be repeated with different runs or different treatments or different hemispheres or even different subjects, assuming the functional relationship among ROIs is fixed during the same processing (i.e., face processing or object processing).

### Reporting summary
Further information on research design is available in the Nature Portfolio Reporting Summary linked to this article.

## Data availability
The data that support the findings of this study are available from the corresponding author, upon request. Data are still being analyzed for other purposes and cannot be made publicly available at this time. Source data are provided with this paper.

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

## Acknowledgements

This paper is dedicated to the memory of our dear colleague and friend, Leslie G. Ungerleider. We thank Roger B.H. Tootell for discussing the experiment design, S. William Li and Katherine B. Jones for the animal training, and Frank Q. Ye, Charles C. Zhu, and David C. Ide for technical assistance. This work was supported by the National Institute of Mental Health Intramural Research Program (ZIAMH002918 to L.G.U.), Science and Technology Innovation 2030 - Brain Science and Brain-Inspired Intelligence Project (Grant No. 2021ZD0200203), Strategic Priority Research Program of Chinese Academy of Science (XDB32020207 to N.L.), the National Natural Science Foundation of China (Grant No. 32071094 to N.L.), and a grant from the National Eye Institute (R01EY027018 to M.B.). M.B. acknowledges support from P30 CORE award EY08098 from the National Eye Institute, NIH, and unrestricted supporting funds from The Research to Prevent Blindness Inc, NY, and the Eye & Ear Foundation of Pittsburgh.

## Author contributions

N.L., F.H.-B., and L.G.U. designed the research; N.L. and J.N.T. performed the research; N.L., M.B., and G.A. analyzed data; and N.L., M.B., G.A., F.H.-B., and L.G.U. wrote the paper.

## Competing interests

The authors declare no competing interests.
