## [Peer Review File · Nature Communications]

Bidirectional and parallel relationships in macaque face circuit revealed by fMRI and causal pharmacological inactivationREVIEWER COMMENTS

Reviewer #1 (Remarks to the Author):

1. Summary

The authors investigated relationship between face-selective patches in the macaque IT cortex in terms of inter-areal information flow. To reveal such relationship in a functional and causal manner, they conducted pharmacological inactivation of each face-selective patch (ML, MF, AL, and AF) while measuring BOLD responses to face or object stimuli (fMRI) in these patches. The following are the main claims from the experimental results:

- 1) Activation of anterior patches requires activation of middle patches.
- 2) Face-selectivity in middle patches requires activation of anterior patches.
- 3) Processing in lateral areas (ML and AL) and that in fundus areas (MF and AF) are relatively independent.

2. Evaluation

The authors have addressed important questions on the functional and causal relationships between macaque face patches, which has not yet been clarified previously and will be crucial for understanding the computational mechanism underlying face-processing and potentially general object processing. The method seems sound, albeit quite conventional.

However, despite the interesting topic, I have to recommend rejection of the present submission, the main reasons being: (A) insufficient justification of claim 2, and (B) poor preparation of the manuscript.

(A) Insufficient justification of claim 2

Claim 2 is the most important among the three since the other two claims are less surprising. However, the experimental results for this seem rather shaky. Specifically, in Section 2.1, it is described that combined AF and AL inactivation reduces responses to faces but not to objects, while individual AF or AL inactivation reduces both responses. This effect seems to be not consistent across hemispheres and

monkeys (Figures S5-8), and the authors excuse that these might have been due to potential experimental failure (lines 197-199). Even though the calculated face-selective indices show results that look meaningful, the justification of the claim does not seem to be completely convincing. Therefore the authors need to either weaken the claim or redo the experiment to eliminate the possibility of experimental failure and thus confirm the consistency of the results.

(B) Poor preparation of the manuscript

The manuscript contains a number of serious typographical and organizational problems so that the overall quality does not meet the high standard of the journal.

* Most conspicuously, Section 2 is textually identical to Section 1, except for specifics in each section. Similarly, Section 2.1 is essentially a repetition of Section 2.1 except for specifics. Such organization is too boring for the readers and unacceptable for a decent journal. The authors should reconsider the entire section design.

* The parts related to amygdala seem too short and poorly motivated (lines 154-155, 206-207) and relevance to what is written in the introduction (lines 78-89) is unclear. If the authors think that these are important enough to be included in the main text, these should be substantially expanded; otherwise, these should be removed from the manuscript.

* Some references are broken: Pitcher et al, Hoffman et al, and Rutishauser et al.

* The method description on the regression analysis used in Section II is missing, so that the results cannot be assessed precisely.

3. Other major comments

In Introduction, a "contrast" is emphasized in line 70. However, this contrast seems too obvious since the anatomical study uses a method that is unable to detect such a serial, hierarchical relationship from the first place. So, such contrast seems not so fair to be criticized as an "inconsistency" (Abstract). The authors should come up with some other story for introducing the motivation.

I suggest a few additional references below to be cited and discussed in the manuscript. Ref 1 is a causal study using a pharmacological inactivation to investigate necessity of ML for face recognition. Ref 2 is about a top-down processing from anterior to middle patches. Ref 3 is about a computational model that predicts the role of a top-down processing for face-selectivity in ML, which accords Claim 2 in the present study.

1. Sadagopan, S., Zarco, W. & Freiwald, W. A. A causal relationship between face-patch activity and face-detection behavior. *Elife* (2017).
2. Schwiedrzik, C. M. & Freiwald, W. A. High-Level Prediction Signals in a Low-Level Area of the Macaque Face-Processing Hierarchy. *Neuron* 96, 89-97.e5 (2017).
3. Hosoya, H. & Hyvärinen, A. A mixture of sparse coding models explaining properties of face neurons related to holistic and parts-based processing. *PLoS Comput. Biol.* 13, e1005667 (2017).

4. Minor comments

exam -> examine (line 200)

evince -> evidence (line 268)

dprime seems wrong (line 541)

Figure S1-8. Some bars are empty and very confusing. This has to be explained.

Reviewer #2 (Remarks to the Author):

Liu et al., in a series of carefully performed experiments study the effect of inactivation of various parts of the inferior temporal face network on the rest of it. The topic and motivation of the study are very important as they reflect an ongoing metamorphosis in our understanding of cortical organization in high level visual areas. I congratulate the authors on their systematic methodical approach to such a fundamental problem.

While I enjoyed very much reading this paper, and while I find the study potentially suitable for publication in *Nature Communications*, I have one central problem with the current manuscript. Below, I will break this problem into two related major concerns followed by my minor concerns.

In general, I find this study potentially important and influential and hope to see the authors' satisfying responses to my major concerns. I am also honored to be reviewing perhaps one of the last to-be-published papers of Leslie Ungerleider and I will do my best to keep the review quality up to her standard.

Major Concerns

1) The amount of injected muscimol ranges from 2.7 to 3.75 μ L. That is a lot of muscimol and IT cortex is packed and small. If the locally injected muscimol bleeds over to the neighboring cortical areas, or worse, to the neighboring face patches it will be a lethal blow to the elegant arguments developed in the manuscript about connectivity. In such case, inactivation of other face patches (or at least a component of the effect) would be the result of direct inactivation by muscimol due to diffusion leakage. That would be consistent with the observed stronger within-hemisphere effects.

I believe the paper needs to go a long way to falsify this possibility in an exhaustive manner. For example, a matrix of physical distances between the cortical injection points can be compared to the matrix of fMRI signal modulations at those sites. Studies of how muscimol spreads in the cortex (Arikan et al. 2002, Allen et al., 2008) need to be mentioned and a proper quantitative discussion regarding this issue needs to be added to the paper.

2) There is no mention of IT cortical areas that are not face selective. Face network cannot be studied in isolation, the curious reader wants to know what happened to the face and object responses of the rest of IT cortex when each of the face patches are inactivated. The results of parts of IT that are not face selective (at least as one lump, ideally as separate lumps each neighboring a face patch) needs to be added to the main plots of the paper and properly discussed.

More critically, the “not face selective” cortical areas between the face patches need to be explored. It is very important to show that remote neural responses are not affected directly by the muscimol and one way of doing it is to measure how muscimol inactivation fades away with distance from the injection site (a functional measure of the muscimol spread function?). Or it might be possible to show that there exist bands of cortex between the face patches that are not altered. Otherwise, the muscimol leakage interpretation gains ground.

In addition, it would be very helpful to present in a main figure, a (or a few) continuous map of functional responses across the cortex (face selective and else) after muscimol injection (with the injection site marked).

Minor Concerns

1) The word “placebo” typically refers to an inactive substance that can affect an outcome by altering the psychological state of the subject/patient (placebo effect). In this case the animal’s cognition is not a concern, so the term placebo is a little misleading. I would use “control” or a word of that quality instead.

2) Muscimol inactivation of the middle face patch is reported to induce behavioral deficits in face recognition only in the contralateral (and not in the ipsilateral) hemifield (Afraz et al. 2015), this is consistent with the observed hemifield effect and might help the discussion against the muscimol leakage interpretation of the results.

3) Muscimol injection rate should be presented in the methods. Also the timing between the injection and scanning should be clarified. This is an important factor in estimating how far muscimol has gone by the time scanning.

Reviewer #3 (Remarks to the Author):

This paper investigates a subset of the face patch circuitry in macaques, using reversible inactivation and fMRI. The approach and the topic are important and timely, and the results, when taken at face value, are interesting and novel. Hence, the study has the potential to be a significant contribution to the understanding of face processing in primates. But in its current form, the presentation of results and methods is insufficiently clear. Additional work on this need to be done to make it a more convincing paper, as suggested below.

Introduction

The paper is well written, but in the introduction, the reasoning seems at times superficial. For instance, the authors contrast the progression in complexity and invariance of face patch responses along the posterior-to-anterior axis with the extensive and largely bidirectional connectivity of the face patch system, but I am not convinced that there is an inherent tension between these findings. We know that the brain is highly interconnected, and most cortical regions belonging to a certain functional circuit (not

just the face patch system) are connected directly and also via subcortical routes. This does not prevent each region to exhibit specific properties. In fact, if we jump ahead, the own conclusions of this study suggest nearly fully bidirectional interconnected connectivity (cf. Figure 10).

In other words, I don't think that "anatomical connections of face patches studied by different approaches in monkeys suggest that there is no clear serial, hierarchical organization of the face-processing network (Grimaldi et al., 2016; Moeller et al., 2008)" – and, by the way, the electrical microstimulation studies do not only reveal direct monosynaptic connectivity (see for instance a recent relevant review "Combining brain perturbation and neuroimaging in non-human primates", <https://www.sciencedirect.com/science/article/pii/S1053811921002949>, for the related discussion). I am not arguing that the rationale of systematically dissecting the causal contribution of each node in the circuit does not have a merit – it definitely does. I am merely suggesting to tone down the apparent (in my opinion, unnecessary contrived) contradiction between the two bodies of findings the authors refer to.

Beyond the face patch system, this is one of the few studies so far that combined fMRI and reversible inactivation in macaques. This is indeed a commendable advance, but the previous work that employed this approach, and the associated interpretational issues, should be mentioned, in the Introduction or in the Discussion. I recommend the again to refer to the same recent review (<https://www.sciencedirect.com/science/article/pii/S1053811921002949>, section 3, Reversible lesions and neuroimaging), and the studies cited therein.

Results

Figure 1. Please show all 4 hemispheres, not just two. I also suggest to add individual surface reconstructions (and perhaps example coronal sections) for each monkey, rather than only showing projections on the template brain. Some of the "patches" in Figure 1B,C do not look to be convincingly separate from each other, which might be due to imperfections in template alignment and projection on a template surface.

Please comment on the absence of the AM patch.

Please comment on why there seems a statistically significant, different from zero, activation (normalized beta coefficient) in the regions that are being inactivated, after the inactivation. Is this expected? Of course, the inactivation is not absolute, but given that the activity for each ROI is extracted from the small 3 mm radius volume around the peak activation voxel that was targeted as the injection

center, and the volume of the injection, I would have expected a complete abolishment of the activity in the few mm sphere around the locus of the injection. Also, what about the Gd co-injection and its influence on the fMRI activity estimates (please see the comment below). All in all, the quantification and the visualization of the injection extent, and its relationship to the activity in the vicinity of the injection, needs to be provided (please see the comment below).

“Results from individual animals and hemispheres showed slight variations from the common pattern which is described below (for additional details, see Figure S1-8).” - Generally, the supplemental figures, consisting of a huge number of bar plots, separately for each monkey, make it very difficult to bring the results together, and to assess the similarity/difference between the individual patterns. Please find another way to plot the data (e.g. using line or symbol plots rather than space-inefficient bar plots), so that both monkeys are plotted on the same figure, for more direct comparison. I also have a suggestion below regarding the possibility to link each main figure result to the schematic in Figure 10.

Figure 10: It is a very crucial figure that is intended to summarize the less digestible collection of multiple bar plots and stats in the previous figures, but it is confusing. What are dashed blue lines? What is the difference between bidirectional red arrows between e.g. ML and MF, and two separate unidirectional red arrows e.g. between AL and ML?

Besides clarifying these aspects, I suggest to use a similar small schematic inset in each of the (ipsilateral) bar panels to graphically indicate which node is inactivated and how (which arrows) it affects, to bring all these results together in a more easily interpretable manner. Otherwise, I had a difficulty relating the specific effects and stats in each main figure (as well as the level of consistency across animals that can be in principle gleaned from the supplemental figures) to the overall conclusions.

Finally, in few cases where the same patch was inactivated in the left and the right hemisphere, in the same animal, was there any difference between inactivation effects in each hemisphere (i.e. indications of hemispheric asymmetry) – or no clear pattern emerged (again, I know the information is distributed across several suppl. figures, but perhaps the authors could succinctly summarize the findings)?

Methods

This is a study with fairly advanced and involved methodology (especially given the nonstandard 4.7T Bruker scanner), but the methodological details are not appropriately described and presented in the figures.

Imaging

Given the high field strength that causes image distortion, please show the raw coronal EPIs for each animal, and corresponding T1w slices. Was an EPI distortion correction methodology used? If yes, which one?

“Target coordinates were confirmed in a subsequent scan session with local infusions of sterile dilute gadolinium (Gd) contrast agent (2.7~3.75 μ L of a 5 mM solution, Asthagiri et al., 2011); this contrast agent was also included for each of the functional injection scanning sessions to visualize the injection spread.”

Figure 1: “The major borders of the inactivation sites were evaluated by coinjected gadolinium, which could be visualized in post-injection anatomical scans, and indicated by the black lines.”

– In the supplement, please show example session scans with Gd, for each face patch, and analyze the spread of Gd across sessions (e.g. using across-session heatmaps). By the way, the reference Asthagiri et al., 2011 is missing in the Reference list.

Inactivation

How was the injection performed? Outside of the scanner, using metal cannulae? Inside, using MRI-compatible plastic cannulae? How long after the injection the data collection started? At what point the Gd injection spread scan was collected?

If the Gd was injected in each inactivation session, how reliable are the estimates of fMRI activity in the inactivated patch?

Table 1 shows, besides reasonable (at least based on the previous fMRI-inactivation work), 5-6 sessions per site, some very low numbers (1-2 injection sessions). Why so few sessions were conducted? Is this a concern regarding the variability of the observed patterns? I honestly tried but failed within reasonable time to obtain a comprehensive picture of consistent and inconsistent effects in each animal, for each experimental condition, by switching back and forth between the Table 1, the main figures (which combine both monkeys, who contribute very different amount of data / runs), and the supplemental figures.

fMRI analysis

“To define ROIs, all runs were concatenated across the initial localizer and placebo sessions, respectively”. Strictly speaking, this approach suffers a bit from a circularity pitfall – since you use “placebo” sessions for defining ROIs. In the absence of a real inactivation effect, the activity during the placebo condition – used to define peak voxel and the volume around it - would be somewhat higher than during the inactivation, just due to selection bias – you select the voxels more active – by chance – during the placebo condition than during the inactivation. I personally do not think it is a major problem, and fairly convinced that if the ROIs were defined purely based on the independent (e.g. localizer-only) dataset the results would be very similar, but the authors should comment on this.

Discussion

One topic that is conspicuously missing from the discussion is the functional significance (for the organism as a whole) of the activations in the contralateral hemisphere, and any interpretation regarding the lack of a consistent inactivation effect in the contralateral hemisphere. Note that here the faces are presented foveally (and arguably are typically viewed foveally in the natural environment). So if one hemisphere is inactivated, what functional consequences should this have if there is another intact hemisphere that can take care of the processing? Are the left and right face patch systems fully redundant, when the stimuli are not lateralized? Is there a strong evidence for the interhemispheric transcallosal connectivity between the left and right face patches, as it has been shown more generally for IT cortex? On the other hand, if such connectivity exists (as anatomical and microstimulation studies indicate), why the inactivation effect is so small and inconsistent in the contralateral hemisphere?

Minor points:

Abstract

“Although the locations of face patches have been well established in primate visual cortex, the functional and anatomical findings reveal inconsistencies in the organization of face patches in macaque cortex.” – “Inconsistencies” in the actual organization, or inconsistencies between functional and anatomical findings? Please clarify this sentence.

“In addition, the above-mentioned studies using electrical microstimulation or retrograde tracers have seldomly targeted patches within the STS fundus, a limit that might have prevented uncovering the presence of a fundal-lateral axis within the IT face network.” – a limitation, rather than “a limit”?

REVIEWER COMMENTS

Below we provide a point-by-point response to the comments of the Reviewers. The Reviewer's comments are in black and our responses are in red. We enclose the revised manuscript showing all changes in the text file with track changes.

Reviewer #1 (Remarks to the Author):

1. Summary

The authors investigated relationship between face-selective patches in the macaque IT cortex in terms of inter-areal information flow. To reveal such relationship in a functional and causal manner, they conducted pharmacological inactivation of each face-selective patch (ML, MF, AL, and AF) while measuring BOLD responses to face or object stimuli (fMRI) in these patches. The following are the main claims from the experimental results:

- 1) Activation of anterior patches requires activation of middle patches.
- 2) Face-selectivity in middle patches requires activation of anterior patches.
- 3) Processing in lateral areas (ML and AL) and that in fundus areas (MF and AF) are relatively independent.

2. Evaluation

The authors have addressed important questions on the functional and causal relationships between macaque face patches, which has not yet been clarified previously and will be crucial for understanding the computational mechanism underlying face-processing and potentially general object processing.

However, despite the interesting topic, I have to recommend rejection of the present submission, the main reasons being: (A) insufficient justification of claim 2, and (B) poor preparation of the manuscript.

>> We appreciate the valuable comments from the reviewer. In the revision, we have incorporated all the comments/corrections suggested by the reviewer. We hope that this sufficiently addresses all concerns raised. We provide details below.

(A) Insufficient justification of claim 2

Claim 2 is the most important among the three since the other two claims are less surprising. However, the experimental results for this seem rather shaky. Specifically, in Section 2.1, it is described that combined AF and AL inactivation reduces responses to faces but not to objects, while individual AF or AL inactivation reduces both responses. This effect seems to be not consistent across hemispheres and monkeys (Figures S5-8), and the authors excuse that these might have been due to potential experimental failure (lines 197-199). Even though the calculated face-selective indices show results that look meaningful, the justification of the claim does not seem to be completely convincing. Therefore the authors need to either weaken the claim or redo the experiment to eliminate the possibility of experimental failure and thus confirm the consistency of the results.

>> We appreciate the reviewer's comments about claim 2.

We noted that, in contrast with combined AF and AL inactivation, inactivation of AF or AL alone (especially AL) also reduced responses to objects in MF and ML. The small sample size of runs in inactivation of AF or AL alone (especially AL) might have induced variability in the observed patterns (combined AF and AL inactivation: n=68; AF alone: n=43; AL alone: n=26), undermining the ability to reach a clear conclusion. Unfortunately, we are unable to redo the experiment for logistic reasons. Adopting the reviewer's suggestion, however, we have moderated our claim in the revised *Discussion*, as follows: "We found that, inactivation of the anterior face patches mainly affected the responses to faces but, only minimally, if at all, the responses to objects in the middle face patches." We have also added this information into the revised Figures 4&10.

Note that inactivation of the anterior face patches might have less influence on the responses to objects than the responses to faces. Thus, we calculated face selectivity indexes under inactivation conditions and compared them with those under the control condition. The variance in the responses to faces and objects across runs, sessions, hemispheres, and monkeys was also taken into account in calculating the face selectivity index (see the Eq.1). To limit the effects of a small sample size of sessions, we conducted a bootstrap resampling (n=10,000) of the run set, which simulates distributions of standardized coefficients if the experiment were to be repeated with different runs or different treatments or different hemispheres or even different subjects. We did

find that inactivation of AF alone significantly modulated the face-selective index in MF, with a similar trend in the modulation effect of inactivation of AL on ML. These findings suggested that the face selectivity in the middle face patches arises, in part, from top-down input from the anterior face patches. We have made these points clearer in the revised *Results*.

$$\text{Face Selectivity Index} = \frac{\mu_{\text{faces}} - \mu_{\text{objects}}}{\sqrt{(\sigma_{\text{faces}}^2 + \sigma_{\text{objects}}^2)/2}} \quad \text{Eq.1}$$

(B) Poor preparation of the manuscript

The manuscript contains a number of serious typographical and organizational problems so that the overall quality does not meet the high standard of the journal.

* Most conspicuously, Section 2 is textually identical to Section 1, except for specifics in each section. Similarly, Section 2.1 is essentially a repetition of Section 2.1 except for specifics. Such organization is too boring for the readers and unacceptable for a decent journal. The authors should reconsider the entire section design.

>> We appreciate the reviewer's comments to improve the manuscript. We have reorganized and rewritten the manuscript, following the suggestions of the reviewer for Sections 1 and 2.

* The parts related to amygdala seem too short and poorly motivated (lines 154-155, 206-207) and relevance to what is written in the introduction (lines 78-89) is unclear. If the authors think that these are important enough to be included in the main text, these should be substantially expanded; otherwise, these should be removed from the manuscript.

>> We appreciate this valuable advice. The relationship between the amygdala and the face patches is crucial for us to understand the potential functional differences between the fundal pathway and the lateral pathway. The amygdala is one of three subcortical brain structures consistently connected with face patches (Moeller et al., 2008; Grimaldi et al., 2016) and exhibits functional relationship with face patches (Hadj-Bouziane et al, 2012). As such, exploring the consequences of face patch inactivations beyond the face network would shed further light on the circuitry and representations. We have made these points clearer in the revised *Introduction* and have also expanded the corresponding parts in the revised *Results*. We also discuss the amygdala findings in the *Discussion* section.

* Some references are broken: Pitcher et al, Hoffman et al, and Rutishauser et al.

>> We thank the reviewer for alerting us to this issue. These references have been added in the revised manuscript.

1. Pitcher, D., Japee, S., Rauth, L. & Ungerleider, X.G. The Superior Temporal Sulcus Is Causally Connected to the Amygdala: A Combined TBS-fMRI Study. *Journal of Neuroscience* **37**, 1156-1161 (2017).
2. Hoffman, K.L., Gothard, K.M., Schmid, M.C. & Logothetis, N.K. Facial-expression and gaze-selective responses in the monkey amygdala. *Current Biology* **17**, 766-772 (2007).
3. Rutishauser, U., Mamelak, A.N. & Adolphs, R. The primate amygdala in social perception - insights from electrophysiological recordings and stimulation. *Trends Neurosci* **38**, 295-306 (2015).

* The method description on the regression analysis used in Section II is missing, so that the results cannot be assessed precisely.

>> We apologize for the incomplete information. We modeled the relationship between one ROI and other ROIs by fitting them into a linear equation (Eq.2). For example, the relationship between AF with AL, MF, and ML when subjects were viewing faces was obtained by

$$Y = a + b_1X_1 + b_2X_2 + b_3X_3 + \varepsilon \quad \text{Eq.2}$$

Where Y is the responses to faces in AF, X₁ is the responses to faces in AL, X₂ is the responses to faces in MF, and X₃ is the responses to faces in ML. All the runs across all the treatments ([previously labelled ‘placebo’] control, combined AF and AL inactivation, AF alone inactivation, AL alone inactivation, combined MF and ML inactivation, MF alone inactivation, ML alone inactivation) and across two hemispheres from both monkeys were used in the model (n=575, see Table S1). To assess differences between standardized coefficients (b₁, b₂, b₃ in the above equation), we conducted a bootstrap resampling (n=10,000) of the run set, which simulates distributions of standardized coefficients if the experiment were to be repeated with different runs or different treatments or different hemispheres or even different subjects, assuming the functional relationship among ROIs is fixed during the same processing (i.e., face processing or object

processing). We have added the corresponding information to the revised *Methods* (section “*Relationship among face patches and the amygdala*”).

3. Other major comments

In Introduction, a "contrast" is emphasized in line 70. However, this contrast seems too obvious since the anatomical study uses a method that is unable to detect such a serial, hierarchical relationship from the first place. So, such contrast seems not so fair to be criticized as an "inconsistency" (Abstract). The authors should come up with some other story for introducing the motivation.

>> We agree with this point. We have revised the *Introduction* to focus the narrative on exploring the functional and causal relationships among the face patches by taking the advantage of the approach in which fMRI and pharmacological inactivation are combined.

I suggest a few additional references below to be cited and discussed in the manuscript. Ref 1 is a causal study using a pharmacological inactivation to investigate necessity of ML for face recognition. Ref 2 is about a top-down processing from anterior to middle patches. Ref 3 is about a computational model that predicts the role of a top-down processing for face-selectivity in ML, which accords Claim 2 in the present study.

1. Sadagopan, S., Zarco, W. & Freiwald, W. A. A causal relationship between face-patch activity and face-detection behavior. *Elife* (2017).
2. Schwiedrzik, C. M. & Freiwald, W. A. High-Level Prediction Signals in a Low-Level Area of the Macaque Face-Processing Hierarchy. *Neuron* 96, 89-97.e5 (2017).
3. Hosoya, H. & Hyvärinen, A. A mixture of sparse coding models explaining properties of face neurons related to holistic and parts-based processing. *PLoS Comput. Biol.* 13, e1005667 (2017).

>> We thank the reviewer for these useful references and have now included them in the appropriate position in the revised manuscript.

4. Minor comments

exam -> examine (line 200)

evince -> evidence (line 268)

dprime seems wrong (line 541)

>> We appreciate the reviewer noting these errors. All of them have been fixed in the revised manuscript.

Figure S1-8. Some bars are empty and very confusing. This has to be explained.

>> We thank the reviewer for this helpful comment and have added clarification in the revised manuscript. An empty bar above a particular histogram means that there was no significant difference between this histogram and the histogram to its left with the same color as the bar, no matter whether multiple comparison corrections were done or not. We include this to indicate to the readers that we had, in fact, compared the data and there was no difference. We have added this information into the revised figure legends.

Reviewer #2 (Remarks to the Author):

Liu et al., in a series of carefully performed experiments study the effect of inactivation of various parts of the inferior temporal face network on the rest of it. The topic and motivation of the study are very important as they reflect an ongoing metamorphosis in our understanding of cortical organization in high level visual areas. I congratulate the authors on their systematic methodical approach to such a fundamental problem.

While I enjoyed very much reading this paper, and while I find the study potentially suitable for publication in Nature Communications, I have one central problem with the current manuscript. Below, I will break this problem into two related major concerns followed by my minor concerns.

In general, I find this study potentially important and influential and hope to see the authors' satisfying responses to my major concerns. I am also honored to be reviewing perhaps one of the last to-be-published papers of Leslie Ungerleider and I will do my best to keep the review quality up to her standard.

>> We are grateful to receive the positive and valuable comments from the reviewer. In the revision, we have incorporated all the comments/corrections suggested by the reviewer. We hope that these responses sufficiently address all concerns raised. We also take seriously your comments about keeping up the quality of the manuscript as a tribute to Leslie and so we hope our revision substantially improves this paper. Thank you for your comments.

Major Concerns

1) The amount of injected muscimol ranges from 2.7 to 3.75 μl . That is a lot of muscimol and IT cortex is packed and small. If the locally injected muscimol bleeds over to the neighboring cortical areas, or worse, to the neighboring face patches it will be a lethal blow to the elegant arguments developed in the manuscript about connectivity. In such case, inactivation of other face patches (or at least a component of the effect) would be the result of direct inactivation by muscimol due to diffusion leakage. That would be consistent with the observed stronger within-hemisphere effects.

I believe the paper needs to go a long way to falsify this possibility in an exhaustive manner. For example, a matrix of physical distances between the cortical injection points can be compared to the matrix of fMRI signal modulations at those sites. Studies of how muscimol spreads in the cortex (Arikan et al. 2002, Allen et al., 2008) need to be mentioned and a proper quantitative discussion regarding this issue needs to be added to the paper.

>> We thank the reviewer for pointing it out. We agree with the reviewer that the spread of muscimol is an important factor that might have affected the precision of the findings in the present study. Therefore, prior to conducting the study, we did additional research and conducted preliminary studies to select the dosage.

The two studies mentioned by the reviewer (Arikan et al. 2002 and Allen et al., 2008) were conducted in rodents. Arikan et al. 2002 tested the effective spread of focally injected muscimol in the cerebellum of rats. 1 μl of 2% muscimol in Ringer's solution was injected. According to their results, in the first 1.5 h, the affected areas were mainly within a radius of 1.5 mm (2 voxels away from the injected voxel) around the center of the injection (see Reply Figure [Figure R] 1). Allen et al., 2008 tested the fluorophore-conjugated muscimol molecule. The injected volumes were 0.5 μl for the basolateral amygdala (BLA) and dorsomedial prefrontal cortex (dmPFC), and

1 μ l for perirhinal cortex (PR). Histological analysis showed that the region of fluorescence was restricted to 0.5–1 mm from the injection site.

Figure R1. from Arikani et al. (2008) showing the spread of 1 μ l of muscimol as a function of time.

Figure R2. from Turchi et al. (2018) showing the positions of Ch4al and Ch4am (A&B) and distinct cortical regions influenced by inactivation of Ch4am and Ch4al (C).

Since the size of the macaque brain (wt. \approx 87.35g) is about 48 times that of the rat brain (wt. \approx 1.802g), in most of the macaque studies, a larger quantity of muscimol is typically injected than that in rodents (e.g., Wellman et al., 2005; Turchi et al., 2018). In Wellman et al., 2005, which is the paper many macaque muscimol studies use as a standard, 18 nmol of muscimol (the same concentration as that used in the present study) and 1 μ l was infused into the amygdala. According to their results, even if the injection target was shifted just by 1mm, they obtained completely different effects. Based on this variability, muscimol may render its effects in a very circumscribed fashion. It is incontrovertible that infusions into different brain regions may result in different diffusion gradients but this heuristic is generally accepted. In Turchi et al., 2018, 1.8 μ l-2.4 μ l muscimol was injected into two small subregions of the nucleus basalis of Meynert (NBM), namely, Ch4am and Ch4al, in macaques. As seen in Figure R2A&B, these two subregions were close to each other. A key finding was that distinct cortical areas were influenced by the inactivation of Ch4am versus Ch4al (Figure R2C). These results indicate that 2.4 μ l muscimol still renders its effects in a relatively circumscribed fashion. Moreover, in one previous face patch inactivation study in macaques (Sadagopan et al., 2017), 5 μ l of muscimol was infused into ML. This amount of muscimol covered most but not all of ML: assuming concentric spheres for both volumes, the theoretical maximum of injection volume contained within ML is 63.8%.

Figure R3. from Talbot et al. 2012 showing the ideal (A) and actual (B) spread of 2.7 μ l muscimol.

In the present study, the amount of injected muscimol ranges from 2.7 to 3.75 μ l, which is comparable to (even lower than) the dosages used in previous studies in monkeys (Turchi et al., 2018; Sadagopan et al., 2017). Before the present study, we carefully tested the extent of muscimol spread in order to choose a reasonable dosage. The current dosages were chosen based on these

preliminary explorations (see Talbot et al., 2012, Figure R3) to yield effective inactivation of face patches but with little inactivation of surrounding regions. In our study, we specifically co-infused the MR contrast agent gadolinium (Gd) with muscimol to closely assess the extent of the cortical spread of the inactivation (Heiss et al., 2010; Wilke et al., 2012). That is, the extent of muscimol spread was then detectable on the anatomical MR scan (T1w) and the Gd signal was used to demarcate the spread of muscimol. We have included additional details and tables (Tables S4&5) with the estimated spread of muscimol diffusion within face patches in the revised version of the manuscript (see the revised *Methods* section “*Local targeting*” and “*Transient inactivation*”, Tables S4&5).

Given the differences in spatial resolution between anatomical images ($1.5 \times 0.5 \times 0.5$ mm) and functional images (1.5 mm isotropic) and inter-session differences (MRI coil placement and contrast agent clearance), rather than computing physical distances, we opted for computing the overlap between the muscimol injection (measured with Gd-signal) and ROIs as described in Sadagopan et al., 2017. On average, we found that, of the total injection volume, 81.74% on average (MF: 88.93%; ML: 85.87%; AF: 79.25%; AL: 73.54%), a value comparable to previous studies (e.g., 76% in Sadagopan et al., 2017), was contained within the targeted face-selective ROIs. Furthermore, the differences in results influenced by the inactivation of patches in the fundus of STS versus in the lateral portions of STS, which were close to each other, also indicated that the injected muscimol was mainly located in the targeted ROIs. For example, the results of MF inactivation were different from those of ML inactivation (Figure 2A), reflecting that the injected muscimol rarely leaked or ‘bled over’ to the neighboring face patches.

Taken together, in the present study, the injected muscimol primarily silenced the targeted face patches but left the surrounding regions less (if at all) affected. That is, the potential muscimol leakage is thought to be minimal and should not affect our major results. We have provided additional information about this concern in the revised *Methods* (section “*Transient inactivation*”) and *Discussion*.

2) There is no mention of IT cortical areas that are not face selective. Face network cannot be studied in isolation, the curious reader wants to know what happened to the face and object responses of the rest of IT cortex when each of the face patches are inactivated. The results of parts

of IT that are not face selective (at least as one lump, ideally as separate lumps each neighboring a face patch) needs to be added to the main plots of the paper and properly discussed.

More critically, the “not face selective” cortical areas between the face patches need to be explored. It is very important to show that remote neural responses are not affected directly by the muscimol and one way of doing it is to measure how muscimol inactivation fades away with distance from the injection site (a functional measure of the muscimol spread function?). Or it might be possible to show that there exist bands of cortex between the face patches that are not altered. Otherwise, the muscimol leakage interpretation gains ground.

In addition, it would be very helpful to present in a main figure, a (or a few) continuous map of functional responses across the cortex (face selective and else) after muscimol injection (with the injection site marked).

>> We appreciate the reviewer’s excellent suggestion. Since we used faces and objects as stimuli, we have now also localized the object-selective regions in addition to localizing the face patches. As shown in Figure S18&19, the object-selective regions (especially those between the anterior and middle face patches) were less affected by inactivation of face patches, indicating that the specificity of our key findings to the face patches alone. This is consistent with previous anatomical evidence that the IT face patches are specifically interconnected. This is an important control result and we have included it in the revised manuscript too in the *Methods, Results and Discussion* sections.

Minor Concerns

1) The word “placebo” typically refers to an inactive substance that can affect an outcome by altering the psychological state of the subject/patient (placebo effect). In this case the animal’s cognition is not a concern, so the term placebo is a little misleading. I would use “control” or a word of that quality instead.

>> We agree. We have replaced “placebo” with “control” in the revised manuscript as the reviewer suggested.

2) Muscimol inactivation of the middle face patch is reported to induce behavioral deficits in face

recognition only in the contralateral (and not in the ipsilateral) hemifield (Afraz et al. 2015), this is consistent with the observed hemifield effect and might help the discussion against the muscimol leakage interpretation of the results.

>> We thank the reviewer for this suggestion. We have cited Afraz et al., 2015 in the revised manuscript to support our findings, which are consistent with that observation.

3) Muscimol injection rate should be presented in the methods. Also the timing between the injection and scanning should be clarified. This is an important factor in estimating how far muscimol has gone by the time scanning.

>> We made microinfusions at the rate of at 0.18 $\mu\text{l}/\text{min}$ (e.g., for the dosage of 2.7 μl , the injection takes 15 min to complete), which was similar to or even slower than that used in previous studies (e.g., 0.2 $\mu\text{l}/\text{min}$ in Wellman et al., 2005, average rate of 0.4 $\mu\text{l}/\text{min}$ in Sadagopan et al., 2017). After the injection, we waited for 12-15 min before retracting the cannula to avoid any spread of muscimol along the track of the cannula. We have added this information into the *Methods* (section “*Transient inactivation*”) of the revision.

Reviewer #3 (Remarks to the Author):

This paper investigates a subset of the face patch circuitry in macaques, using reversible inactivation and fMRI. The approach and the topic are important and timely, and the results, when taken at face value, are interesting and novel. Hence, the study has the potential to be a significant contribution to the understanding of face processing in primates. But in its current form, the presentation of results and methods is insufficiently clear. Additional work on this need to be done to make it a more convincing paper, as suggested below.

>> We appreciate these valuable comments and constructive advice. In the revision, we have incorporated all the comments/corrections suggested by this reviewer and hope that these changes address all concerns raised.

Introduction

The paper is well written, but in the introduction, the reasoning seems at times superficial. For

instance, the authors contrast the progression in complexity and invariance of face patch responses along the posterior-to-anterior axis with the extensive and largely bidirectional connectivity of the face patch system, but I am not convinced that there is an inherent tension between these findings. We know that the brain is highly interconnected, and most cortical regions belonging to a certain functional circuit (not just the face patch system) are connected directly and also via subcortical routes. This does not prevent each region to exhibit specific properties. In fact, if we jump ahead, the own conclusions of this study suggest nearly fully bidirectional interconnected connectivity (cf. Figure 10).

In other words, I don't think that "anatomical connections of face patches studied by different approaches in monkeys suggest that there is no clear serial, hierarchical organization of the face-processing network (Grimaldi et al., 2016; Moeller et al., 2008)" – and, by the way, the electrical microstimulation studies do not only reveal direct monosynaptic connectivity (see for instance a recent relevant review "Combining brain perturbation and neuroimaging in non-human primates", <https://www.sciencedirect.com/science/article/pii/S1053811921002949>, for the related discussion). I am not arguing that the rationale of systematically dissecting the causal contribution of each node in the circuit does not have a merit – it definitely does. I am merely suggesting to tone down the apparent (in my opinion, unnecessary contrived) contradiction between the two bodies of findings the authors refer to.

>> We appreciate this important point and we totally agree with it. We have reorganized and rewritten the *Introduction* based on this suggestion. In the revision, we primarily focus on exploring the functional and causal relationships among the face patches by taking the advantage of the combination of fMRI and pharmacological inactivation (and, importantly, we do not propose a contradiction between the two bodies of work, anatomical vs. functional).

Beyond the face patch system, this is one of the few studies so far that combined fMRI and reversible inactivation in macaques. This is indeed a commendable advance, but the previous work that employed this approach, and the associated interpretational issues, should be mentioned, in the Introduction or in the Discussion. I recommend the again to refer to the same recent review (<https://www.sciencedirect.com/science/article/pii/S1053811921002949>, section 3, Reversible lesions and neuroimaging), and the studies cited therein.

>> We thank the reviewer for this suggestion. We have added the corresponding information into the *Introduction* and *Discussion*. For example, in the revised *Introduction*, we have added “With the advantage of being repeatable, with interleaved recovery periods, the approach of transient inactivation has proved to be a powerful tool in exploring the functional roles of specific brain areas. This approach has been broadly used in behavioral studies, combined with electrophysiological recordings and, more recently, with fMRI in macaques. This combination permits the investigation of whole-brain activity and provides a unique opportunity to explore the causal relationship among brain areas (for review, see Klink et al., 2021).”

Results

Figure 1. Please show all 4 hemispheres, not just two. I also suggest to add individual surface reconstructions (and perhaps example coronal sections) for each monkey, rather than only showing projections on the template brain. Some of the “patches” in Figure 1B,C do not look to be convincingly separate from each other, which might be due to imperfections in template alignment and projection on a template surface.

>> This is a good point. In the revision, we have created individual surface reconstructions (Figure 1) and added the coronal EPI sections (Figures S12&13) to show the locations of face patches from the initial localizers. We can see clearly separate peaks of each face patches (especially in Figures S12&13), though some of them did appear to be confluent with each other (especially MF and ML). Consistently with previous studies (e.g., Tsao et al., 2008), ML and MF may in particular be confluent in some hemispheres.

Please comment on the absence of the AM patch.

>> In the present study, we did locate the AM patches in Monkey C (Figure S12) but not in Monkey D (Figure S13). AM patch is located on the ventral surface of IT, which is the first and most affected area by the MION accumulation. Therefore, in Monkey C, presumably because of this accumulation, we were unable to localize AM in many later sessions. Moreover, since we could not locate AM in Monkey D even in the initial localizer experiment, we did not include AM in the present study.

For Monkey D, there are two possibilities for the absence of the AM patch. One is the technical issue concerning poor signal in this area and another is the individual differences in the patterns of face patches.

With respect to the technical issue, among the face patches in the temporal cortex (typically 6 discrete face patches distributed along the posterior-anterior axis: PL, MF, ML, AF, AL, AM), the AM face patch is the most anterior and is also located on the ventral surface but not on the STS lip or fundus as others. Due to relatively weaker imaging signals in the anterior temporal lobe (especially the ventral part) compared with other portions of the temporal cortex, AM has been more recently characterized in monkeys and humans (so called ‘ATL’, the face patch located at the anterior pole of the temporal lobes) in humans. In the earlier studies, AM was not reported (e.g., Tsao et al., 2003; Bell et al., 2008).

For the second possible issue, the individual differences in the patterns of face patches may also explain the absence of the AM in the present study. As shown in Tsao et al., 2008 in which the face localizer experiments were conducted on 9 monkeys, individual animals and hemispheres did exhibit slight variations from the typical pattern of 6 face patches. One of the typical variations is that 1 or more (including the AM) of the 6 patches were absent.

We have added the corresponding information into the revised *Methods* (section “*Definition of face-selective ROIs, the amygdala and object-selective regions*”).

Please comment on why there seems a statistically significant, different from zero, activation (normalized beta coefficient) in the regions that are being inactivated, after the inactivation. Is this expected? Of course, the inactivation is not absolute, but given that the activity for each ROI is extracted from the small 3 mm radius volume around the peak activation voxel that was targeted as the injection center, and the volume of the injection, I would have expected a complete abolishment of the activity in the few mm sphere around the locus of the injection.

>> We thank the reviewer for pointing this out. There are two possibilities for non-zero normalized beta coefficient in the inactivated sites.

Firstly, the statistical results were based on the beta coefficients from each run as described in the *Methods*. Though the mean value of the inactivated site might be greater than zero, it should

be noted that not all of these beta coefficients were significantly greater than zero. We have shown the activation maps in the revised manuscript to provide a complete picture of the effects of inactivation. As shown in Figures S18&19, at the selected threshold ($p < 0.005$ uncorrected), there were no, or only a few, voxels that survived this statistical threshold after inactivation (especially within the defined ROIs).

Secondly, while we tried to target the center of the ROIs as accurately as possible, we also tried to avoid the vessels. Therefore, the center of injection might be slightly off from the center of the ROIs. As we note in response to the other reviewers, we analyzed the location of injection based on the signal of Gd and found that, on average across all the ROIs, about 45.39% of ROIs on average (comparable to values [44%] in ML inactivation in Sadagopan et al., 2017) across all the ROIs overlapped with the injection sites. Noted that the theoretical ROI overlap, assuming concentric spheres for both ROIs and injection sites, was 56.08% on average. Since we did not draw the ROI based on the center of the injection site, there might still be signals (but indeed close to 0) in the ROIs defined in the present study.

We have discussed this issue in the *Results* section of the revised manuscript.

Also, what about the Gd co-injection and its influence on the fMRI activity estimates (please see the comment below).

All in all, the quantification and the visualization of the injection extent, and its relationship to the activity in the vicinity of the injection, needs to be provided (please see the comment below).

>> We thank the reviewer for this suggestion.

Because we share the concern of the reviewers on this issue, we examined the research in the literature. Gd co-injections have been used in visualizing the injection sites and the cortical spread of Gd could closely tracks the diffusion extent of muscimol (Heiss et al., 2010; Wilke et al., 2012). Gd mainly affects the T1-weighted signals (25%) but leaves T2-weighted signals less affected (4%) (see details in see <https://mriquestions.com/why-does-gd-shorten-t1.html>). Functional results are based on the T2*-weighted signals and previous studies have used a similar approach in monkey fMRI studies (Wilke et al., 2012; Turchi et al., 2018). Moreover, we tested the potential effects of the Gd-induced T1-weighted signals on the functional activity. When comparing the activation

maps with Saline injection with those with Gd injections, we did not find significant differences. Furthermore, to control for the potential effects of Gd on the fMRI activity (though it may be very weak), in the vehicle ([previously labelled 'placebo'] control in the revision) conditions, we injected the same volume of Gd into the target region as was used for the Gd plus muscimol condition; thus, the differences between control and inactivation sessions should mainly have been caused by muscimol. We have added corresponding information in the revised *Methods* (section “*Local targeting*”).

Moreover, we have analyzed the spread of Gd as previous studies (Sadagopan et al., 2017) and quantified the extent of the injection (Tables S4&5). As stated in the response to the Reviewer #2, we can validate the effect of the muscimol injection and have done so. We have included this point in the revised manuscript.

“Results from individual animals and hemispheres showed slight variations from the common pattern which is described below (for additional details, see Figure S1-8).” - Generally, the supplemental figures, consisting of a huge number of bar plots, separately for each monkey, make it very difficult to bring the results together, and to assess the similarity/difference between the individual patterns. Please find another way to plot the data (e.g. using line or symbol plots rather than space-inefficient bar plots), so that both monkeys are plotted on the same figure, for more direct comparison. I also have a suggestion below regarding the possibility to link each main figure result to the schematic in Figure 10.

>> We thank the reviewer for this advice. We have added two supplemental figures as the reviewer suggested to facilitate more direct comparisons across hemispheres/monkeys (Figures S1&2 in the revision). Since it is difficult to mark the significance of differences between conditions for all the hemispheres in one figure, the original supplemental figures have been also kept (Figures S3-10 in the revision). Moreover, we have linked each main figure (e.g., Figures 2, 4, and 7) to the schematic in Figure 10 by adding the small schematic inset.

Figure 10: It is a very crucial figure that is intended to summarize the less digestible collection of multiple bar plots and stats in the previous figures, but it is confusing. What are dashed blue lines?

What is the difference between bidirectional red arrows between e.g. ML and MF, and two separate unidirectional red arrows e.g. between AL and ML?

>> We appreciate these concerns and have modified this figure accordingly in the revised manuscript. Briefly, the Red line means effects on responses to faces; the Blue line means effects on responses to object; the Black dashed lines means weak or variable effects on responses to both faces and objects; the Blue dashed lines means weak or variable effects on responses to objects. Note that Figure 10 reflects both the results of *“Impact of inactivation on face patches within the IT cortex”* and *“Causal Relationship among the IT face network and the amygdala”*, thus it shows slightly differences from schematic insets in Figures 2&4.

Besides clarifying these aspects, I suggest to use a similar small schematic inset in each of the (ipsilateral) bar panels to graphically indicate which node is inactivated and how (which arrows) it affects, to bring all these results together in a more easily interpretable manner. Otherwise, I had a difficulty relating the specific effects and stats in each main figure (as well as the level of consistency across animals that can be in principle gleaned from the supplemental figures) to the overall conclusions.

>> Thanks for this advice. We have added the small schematic inset into Figures in the revised manuscript as the reviewer suggested. This definitely helps convey the key points.

Finally, in few cases where the same patch was inactivated in the left and the right hemisphere, in the same animal, was there any difference between inactivation effects in each hemisphere (i.e. indications of hemispheric asymmetry) – or no clear pattern emerged (again, I know the information is distributed across several suppl. figures, but perhaps the authors could succinctly summarize the findings)?

>> We appreciate this comment and thank the reviewer. There were no obvious hemispheric asymmetries in the inactivation experiments. Given the lack of consistency and small sample size, we refrain from discussing further any left versus right hemispheric differences in the present manuscript even slight variations were noted. The results from left and right hemispheres were

collapsed in the analyses. We have added the corresponding information into the revised *Results* and *Methods*.

Methods

This is a study with fairly advanced and involved methodology (especially given the nonstandard 4.7T Bruker scanner), but the methodological details are not appropriately described and presented in the figures.

Imaging

Given the high field strength that causes image distortion, please show the raw coronal EPIs for each animal, and corresponding T1w slices. Was an EPI distortion correction methodology used? If yes, which one?

>> As many previous studies conducted at the 4.7T scanner (Liu et al., 2015; Zhang et al., 2020; but see Turchi et al., 2018), the distortion correction methodology was not conducted in the present study. The general distortion did not affect the areas of interest (face patches) in the present study (see Figures S12&13). Therefore, we analyzed the data based on uncorrected EPI images. Since the alignments across sessions were well equated (we benefited from the consistent locations of animals' heads in the scanner), the distortion should not affect the functional results but may slightly affect the visualization of activation maps on the anatomical images and surfaces, which were conducted based on T1w images.

“Target coordinates were confirmed in a subsequent scan session with local infusions of sterile dilute gadolinium (Gd) contrast agent (2.7~3.75 μ L of a 5 mM solution, Asthagiri et al., 2011); this contrast agent was also included for each of the functional injection scanning sessions to visualize the injection spread.”

Figure 1: “The major borders of the inactivation sites were evaluated by coinjected gadolinium, which could be visualized in post-injection anatomical scans, and indicated by the black lines.”

– In the supplement, please show example session scans with Gd, for each face patch, and analyze the spread of Gd across sessions (e.g. using across-session heatmaps).

>> We have added some example sessions with Gd to show the spread of Gd (Figures S14-17). Since the number of sessions under different inactivation sites was limited, the value of heatmaps presenting whether and how many times one voxel may be inactivated by muscimol could not reflect the spread of Gd well. Instead, we calculated volumetric overlaps between the inactivation injections and face patches for each session using similar methods as that of Sadagopan et al., (2017). We have added these results in Tables S4&5.

By the way, the reference Asthagiri et al., 2011 is missing in the Reference list.

>> We are sorry for this oversight. We have now added it into the revised Reference list.

Inactivation

How was the injection performed? Outside of the scanner, using metal cannulae? Inside, using MRI-compatible plastic cannulae? How long after the injection the data collection started? At what point the Gd injection spread scan was collected?

>> The injections were performed outside the scanner with metal cannulas. When the injection was done (15~21 mins depending on the injected volume), we waited 15-20 mins (also depending on the injected volume) before removing the cannula. Then, we removed the injection grid, cleaned the chamber, transferred animals to the scanner room, and started the setup for the scanning. Therefore, the scanning started about 50 min after the injection. To get a clear image of the Gd injection spread, we collected a T1w image first. Then we lowered the animal outside the bore of the vertical scanner (keeping the animal's head fixed and the coil around the animal's head) and did the MION injection, and then raised the animals into the same location of the bore of the scanner. That is, the functional scanning started about 10 mins after the T1w image collection, that is 1 hour after injection (including the waiting time). The fMRI data collection lasted about 1.25 hrs (~ 10 runs). Thus, the fMRI data were acquired between ~1 and 2.5 hrs from the end of the

injection procedure, falling into the window of the maximal muscimol effects (~0.5 hrs to ~4 hrs) (Lomber, 1999; Dias and Segraves, 1999; Sommer and Wurtz, 2004). We have added this information into the revised *Methods* (“*Transient inactivation*”).

If the Gd was injected in each inactivation session, how reliable are the estimates of fMRI activity in the inactivated patch?

>> Please see the reply to “what about the Gd co-injection and its influence on the fMRI activity estimates”.

Table 1 shows, besides reasonable (at least based on the previous fMRI-inactivation work), 5-6 sessions per site, some very low numbers (1-2 injection sessions). Why so few sessions were conducted? Is this a concern regarding the variability of the observed patterns? I honestly tried but failed within reasonable time to obtain a comprehensive picture of consistent and inconsistent effects in each animal, for each experimental condition, by switching back and forth between the Table 1, the main figures (which combine both monkeys, who contribute very different amount of data / runs), and the supplemental figures.

>> We applied the following two strategies: 1) to keep the inactivation sessions and control sessions interleaved (1 control then 2-3 inactivations or so) to limit the effects of periods on animals' status, MION accumulation, and so on; 2) to do the same set of inactivation (e.g., M which including combined MF and ML inactivation, MF alone inactivation, ML alone inactivation) together to perform the comparisons among them and between them within the same control session set. The ideal number is 5-6 as the reviewer noticed. Though we tried our best to control the number of sessions equally across all the treatments, when we conducted the experiments, the final number of sessions still varied. Since some targets were very deep (especially those at the anterior temporal lobe, i.e., AF and AL) and close to vessels (especially those at the fundus, i.e., AF and MF, which had also been less accessible than other targets in previous studies for the same reason [e.g., Grimaldi et al., 2016]), not every injection was successful. Only those instances, in which we confirmed that the injection was successful (reaching the target sites well) and the spread of Gd was ideal (e.g., covering most of the ROIs), were used in the final analyses. For example, on multiple occasions, for the right hemisphere of Monkey D, we tried inactivations of both F and

L as well as F alone but were unable to get these inactivations to work well (e.g., off the targeted ROIs or the cannula was blocked). We considered the problem of MION accumulation and the possibility of reaching the desired target successfully each time and, after several failed attempts, we decided to give up and move on to another injection site. In this case, we did get more control sessions due to the above strategy #1 and we used them all to increase the statistical power. The differences in the number of sessions and runs did cause some variability in the observed individual (hemisphere) patterns. Therefore, the overall results in which all the data from both monkeys and hemispheres were considered together to increase statistical power and reliability.

fMRI analysis

“To define ROIs, all runs were concatenated across the initial localizer and placebo sessions, respectively”. Strictly speaking, this approach suffers a bit from a circularity pitfall – since you use “placebo” sessions for defining ROIs. In the absence of a real inactivation effect, the activity during the placebo condition – used to define peak voxel and the volume around it - would be somewhat higher than during the inactivation, just due to selection bias – you select the voxels more active – by chance – during the placebo condition than during the inactivation. I personally do not think it is a major problem, and fairly convinced that if the ROIs were defined purely based on the independent (e.g. localizer-only) dataset the results would be very similar, but the authors should comment on this.

>> We are sorry for not making this clearer. Briefly, a two-step ROI definition method was adopted in the present study. We agree that, ideally, we should have used the localizer-only dataset to define the ROIs. We did do this at first but because of the MION accumulation, we could not get signals from all the voxels within the initially defined ROIs. If we included those unresponsive voxels in our final analyses, the mean signals would likely be weak even in the control conditions, and then it would be difficult to compare the control conditions with the inactivation conditions. For example, when we conducted inactivations of the left middle face patches in Monkey D, no voxels were obtained at the selected threshold ($p < 0.005$ uncorrected) from the initially-defined LAL and the mean beta coefficients of the initially-defined LAL was close to 0 even in the control conditions. Therefore, we conducted the second step: within the initially-defined ROIs, we excluded those voxels that did not survive in the control conditions. Though this two-step ROI

definition method may not be ideal, the approach to defining ROI should not affect the results substantially. We have made the two-step ROI definition method clearer in the revised *Methods* (section “*Definition of face-selective ROIs, the amygdala and object-selective regions*”).

Discussion

One topic that is conspicuously missing from the discussion is the functional significance (for the organism as a whole) of the activations in the contralateral hemisphere, and any interpretation regarding the lack of a consistent inactivation effect in the contralateral hemisphere. Note that here the faces are presented foveally (and arguably are typically viewed foveally in the natural environment). So if one hemisphere is inactivated, what functional consequences should this have if there is another intact hemisphere that can take care of the processing? Are the left and right face patch systems fully redundant, when the stimuli are not lateralized? Is there a strong evidence for the interhemispheric transcallosal connectivity between the left and right face patches, as it has been shown more generally for IT cortex? On the other hand, if such connectivity exists (as anatomical and microstimulation studies indicate), why the inactivation effect is so small and inconsistent in the contralateral hemisphere?

>> This is a very important issue and the questions raised are all interesting. According to one previous study (Afraz et al., 2015), suppression of the activities of face cells in one hemisphere by optogenetic and pharmacological approaches did not affect (at least not decrease) the activity of the other hemisphere: the inactivation only caused a decrement in face-discrimination performance when the stimuli were presented in the contralateral visual field but not in the ipsilateral visual field. They even found that the performance increased a little, though not significantly, when the stimuli were presented in the ipsilateral visual field. In the present study, as the reviewer noticed, there was no clear pattern for the contralateral hemisphere. Since we did not require animals to perform complex face-related tasks, we, unfortunately, cannot provide any claim regarding the impact of disrupting the face networks in one hemisphere on face processing abilities.

The previous microstimulation study (Moeller et al., 2008) did find activation, which was weaker and not present in all the cases, in face patches contralateral to the stimulation site. Moreover, the tracer study also found weaker connections with face patches in the contralateral

hemisphere to the injection site. Therefore, interhemispheric transcallosal connectivity is likely to exist. However, at this point, it remains unclear what kind of information is transferred through these interhemispheric transcallosal connections. Since, in the present study, the animals did not perform face-related tasks (fixation only), further studies with a well-designed task should be conducted.

We have added this corresponding discussion to the revised manuscript, and there are clearly many questions that remain to be addressed in future research.

Minor points:

Abstract

“Although the locations of face patches have been well established in primate visual cortex, the functional and anatomical findings reveal inconsistencies in the organization of face patches in macaque cortex.” – “Inconsistencies” in the actual organization, or inconsistencies between functional and anatomical findings? Please clarify this sentence.

>> We thank the reviewer for this request and following the comments of Reviewer #1, we have reorganized the narrative, removing this sentence from the revised *Abstract*.

“In addition, the above-mentioned studies using electrical microstimulation or retrograde tracers have seldomly targeted patches within the STS fundus, a limit that might have prevented uncovering the presence of a fundal-lateral axis within the IT face network.” – a limitation, rather than “a limit”?

>> Thanks. We have corrected this in the revision.

Reference:

1. Moeller, S., Freiwald, W.A. & Tsao, D.Y. Patches with links: a unified system for processing faces in the macaque temporal lobe. *Science* **320**, 1355-1359 (2008).
2. Grimaldi, P., Saleem, K.S. & Tsao, D. Anatomical Connections of the Functionally Defined "Face Patches" in the Macaque Monkey. *Neuron* **90**, 1325-1342 (2016).
3. Cao, R.N., Li, X., Brandmeir, N.J. & Wang, S. Encoding of facial features by single neurons in the human amygdala and hippocampus. *Commun Biol* **4**(2021).
4. Pitcher, D., Japee, S., Rauth, L. & Ungerleider, X.G. The Superior Temporal Sulcus Is Causally Connected to the Amygdala: A Combined TBS-fMRI Study. *Journal of Neuroscience* **37**, 1156-1161 (2017).
5. Hoffman, K.L., Gothard, K.M., Schmid, M.C. & Logothetis, N.K. Facial-expression and gaze-selective responses in the monkey amygdala. *Current Biology* **17**, 766-772 (2007).
6. Rutishauser, U., Mamelak, A.N. & Adolphs, R. The primate amygdala in social perception - insights from electrophysiological recordings and stimulation. *Trends Neurosci* **38**, 295-306 (2015).
7. Arikian, R., *et al.* A method to measure the effective spread of focally injected muscimol into the central nervous system with electrophysiology and light microscopy. *J Neurosci Meth* **118**, 51-57 (2002).
8. Allen, T.A., *et al.* Imaging the spread of reversible brain inactivations using fluorescent muscimol. *J Neurosci Meth* **171**, 30-38 (2008).
9. Turchi, J., *et al.* The Basal Forebrain Regulates Global Resting-State fMRI Fluctuations. *Neuron* **97**, 940-952 e944 (2018).
10. Wellman, L.L., Gale, K. & Malkova, L. GABAA-mediated inhibition of basolateral amygdala blocks reward devaluation in macaques. *The Journal of neuroscience : the official journal of the Society for Neuroscience* **25**, 4577-4586 (2005).
11. Sadagopan, S., Zarco, W. & Freiwald, W.A. A causal relationship between face-patch activity and face-detection behavior. *Elife* **6**(2017).
12. Heiss, J.D., Walbridge, S., Asthagiri, A.R. & Lonser, R.R. Image-guided convection-enhanced delivery of muscimol to the primate brain. *J Neurosurg* **112**, 790-795 (2010).

13. Wilke, M., Kagan, I. & Andersen, R.A. Functional imaging reveals rapid reorganization of cortical activity after parietal inactivation in monkeys. *Proceedings of the National Academy of Sciences of the United States of America* **109**, 8274-8279 (2012).
14. Talbot, T., Ide, D., Liu, N. & Turchi, J. A novel, variable angle guide grid for neuronal activity studies. *Frontiers in integrative neuroscience* **6**, 1 (2011).
15. Afraz, A., Boyden, E.S. & DiCarlo, J.J. Optogenetic and pharmacological suppression of spatial clusters of face neurons reveal their causal role in face gender discrimination. *Proceedings of the National Academy of Sciences of the United States of America* **112**, 6730-6735 (2015).
16. Klink, P.C., *et al.* Combining brain perturbation and neuroimaging in non-human primates. *Neuroimage* **235**, 118017 (2021).
17. Tsao, D.Y., Moeller, S. & Freiwald, W.A. Comparing face patch systems in macaques and humans. *Proceedings of the National Academy of Sciences of the United States of America* **105**, 19514-19519 (2008).
18. Tsao, D.Y., Freiwald, W.A., Knutsen, T.A., Mandeville, J.B. & Tootell, R.B. Faces and objects in macaque cerebral cortex. *Nat Neurosci* **6**, 989-995 (2003).
19. Bell, A.H., Hadj-Bouziane, F., Frihauf, J.B., Tootell, R.B. & Ungerleider, L.G. Object representations in the temporal cortex of monkeys and humans as revealed by functional magnetic resonance imaging. *J Neurophysiol* (2008).
20. Liu, N., *et al.* Oxytocin modulates fMRI responses to facial expression in macaques. *Proceedings of the National Academy of Sciences of the United States of America* **112**, E3123-3130 (2015).
21. Zhang, H., Japee, S., Stacy, A., Flessert, M. & Ungerleider, L.G. Anterior superior temporal sulcus is specialized for non-rigid facial motion in both monkeys and humans. *Neuroimage* **218**(2020).
22. Asthagiri, A.R., Walbridge, S., Heiss, J.D. & Lonser, R.R. Effect of concentration on the accuracy of convective imaging distribution of a gadolinium-based surrogate tracer Laboratory investigation. *J Neurosurg* **115**, 467-473 (2011).
23. Lomber, S.G. The advantages and limitations of permanent or reversible deactivation techniques in the assessment of neural function. *J Neurosci Methods* **86**, 109-117 (1999).

24. Dias, E.C. & Segraves, M.A. Muscimol-induced inactivation of monkey frontal eye field: effects on visually and memory-guided saccades. *J Neurophysiol* **81**, 2191-2214 (1999).
25. Sommer, M.A. & Wurtz, R.H. What the brain stem tells the frontal cortex. II. Role of the SC-MD-FEF pathway in corollary discharge. *Journal of Neurophysiology* **91**, 1403-1423 (2004).

REVIEWER COMMENTS

Reviewer #1 (Remarks to the Author):

The authors have made a substantial revision on the initial manuscript. The revision has dealt with all of my previous comments; in particular, the clarity has been tremendously improved. As a result, my concern about insufficient justification is not cleared, which makes me consider that this study now deserves publication in the journal.

However, there are still a few minor clarify/writing problems that need to be dealt with before actual publication.

L. 205-206. Please add more background on why the result on the object-selective regions is important.

L. 209-214. This section introduction sounds a bit too sudden. Please make it more gentle and smooth.

L. 245. It refers to Eq. 1, but it's in a far place in the Methods section. Please say so, or bring the equation here.

L. 273, 274, 289. The term "causal relationship" seems inappropriate since only predictivity between patches is examined. Please replace it with some more modest term. How about "functional relationship"?

Fig. 8. "MF" in the third plot from the left, in the bottom row should be "AL".

Supplementary Materials:

Some figure numbers are confused. The last figure should be labeled "Figure S19"?

Reviewer #2 (Remarks to the Author):

The authors have successfully addressed all of my concerns. I now find the manuscript ready for publication in Nature Communications.

Congratulations on shipping an excellent paper.

Arash Afraz.

Reviewer #3 (Remarks to the Author):

The authors did a very good job addressing most comments of the reviewers in the revision. I only have one remaining point that the authors need to consider before the publication:

As reviewer 2 also pointed out, in the context of this study it is very important to assess the spread of injection to rule out (or quantify) the leakage to neighboring regions. The authors only partially addressed this point, but I think the paper will benefit from more systematic analysis of the overlaps, and some clarifications.

"The volume of injections was defined as all the voxels with values greater than 50% of the maximum value of Gd brightness on T1w images within a radius of 2.8 mm around the injection site." - could this procedure be illustrated, to see what are the the resulting injection borders and the corresponding estimate of the injection volume? This can be added to Figures S14-17, or as additional figures. The order of S14-17 is confused in the Supplement. I would also crop the relevant hemisphere without midline, to zoom in to the relevant parts of the scan, and to avoid image saturation by very bright intensity in S17.

The figures S14-17 are not clear: e.g. what does LML inactivation mean in the context of "From a MF&ML session"? - i.e. why not both are shown to be inactivated? In other words, I expected to see Gd only in MF in MF session, only in ML in ML session, and in both ROIs in MF&ML session.

"The borders of targeted ROIs are indicated by the blue lines, while the borders of nearby non-targeted ROIs are indicated by the green lines." - but in some cases, it appears that there is as much Gd within green ROI as within blue ROI (e.g. S15 LML inactivation middle sections, S16 RAL inactivation left section).

[PREVIOUS COMMENT – In the supplement, please show example session scans with Gd, for each face patch, and analyze the spread of Gd across sessions (e.g. using across-session heatmaps).]

RESPONSE >> We have added some example sessions with Gd to show the spread of Gd (Figures S14-17). Since the number of sessions under different inactivation sites was limited, the value of heatmaps presenting whether and how many times one voxel may be inactivated by muscimol could not reflect the spread of Gd well. Instead, we calculated volumetric overlaps between the inactivation injections and face patches for each session using similar methods as that of Sadagopan et al., (2017). We have added these results in Tables S4&5.

I still think the probability maps can easily be done even across varying number of sessions and is a straightforward way to assess both the extent and reproducibility of the injections. The quantification along the lines of Tables S4 and S5 is also OK - but then it needs some indication about variance across sessions - i.e. mean, SD and range of percent overlap of injection and ROI volumes.

Importantly, I am missing in the Tables S4 and S5 the data on inadvertent (due to leakage) spread of Gd to adjacent ROIs, apparent in the figures. For instance, MF Inactivation row only shows the overlap with MF ROI while the columns related to ML are left empty. Please report the values for the adjacent ROIs (and if they are mostly low, this would be a good argument against the diffusion leakage concern!).

Table S4 is presumably only showing data on middle patches, but the legend says "Volumetric overlap between the inactivation injections in the middle face patches and the __four__ IT face patches" - and likewise, S5 is concerned with anterior patches but says "Volumetric overlap between the inactivation injections in the anterior face patches and the __four__ IT face patches".

Minor:

I was wondering if the same concentration of Gd (1:1200) that was used for chamber grid localization was also used for the intracortical injection, and whether the 1:1200 means the 1 part of the Magnevist solution (0.5 mmol/mL?) and 1200 parts of saline, or this diluted proportion (1:1200) also accounts for the dilution in the Magnevist bottle?

Again, congratulations on the great work, I look forward to see it published.

REVIEWER COMMENTS

Reviewer #1 (Remarks to the Author):

The authors have made a substantial revision on the initial manuscript. The revision has dealt with all of my previous comments; in particular, the clarity has been tremendously improved. As a result, my concern about insufficient justification is not cleared, which makes me consider that this study now deserves publication in the journal.

>> We are grateful to receive the positive and valuable comments from the reviewer.

However, there are still a few minor clarify/writing problems that need to be dealt with before actual publication.

L. 205-206. Please add more background on why the result on the object-selective regions is important.

>> We thank the reviewer for this suggestion. In the temporal cortex, face patches were surrounded by non-face-selective regions (e.g., object-selective regions). Therefore, to understand whether and how the face network connects with surrounding non-face-selective cortex, it is also important to investigate whether the inactivation of faces patches might affect the object-selective regions. Moreover, the results on the object-selective regions would also clarify whether the remote effect of inactivation found in the present study were actually directly caused by the muscimol spread. We have added this information into the revision.

L. 209-214. This section introduction sounds a bit too sudden. Please make it more gentle and smooth.

>> We agree with the reviewer. We have added some transition sentences to make the description of this introduction smoother. Please see lines 230-234.

L. 245. It refers to Eq. 1, but it's in a far place in the Methods section. Please say so, or bring the equation here.

>> We thank the reviewer for this suggestion. We have brought the equation to its proper location in the revision.

L. 273, 274, 289. The term "causal relationship" seems inappropriate since only predictivity between patches is examined. Please replace it with some more modest term. How about "functional relationship"?

>> We thank the reviewer for this advice. We have replaced "causal" with "functional" as the reviewer suggested regarding the results from the regression analyses.

Fig. 8. "MF" in the third plot from the left, in the bottom row should be "AL".

>> Thanks for alerting us to this matter. We have corrected it in the revision.

Supplementary Materials:

Some figure numbers are confused. The last figure should be labeled "Figure S19"?

>> We appreciate the reviewer noting these errors. This has been fixed in the revised supplementary figures section. Moreover, we have changed the order of supplementary figures to reflect their narrative in the manuscript.

Reviewer #2 (Remarks to the Author):

The authors have successfully addressed all of my concerns. I now find the manuscript ready for publication in Nature Communications.

Congratulations on shipping an excellent paper.

>> We are grateful to receive the overall positive comments from the reviewer.

Reviewer #3 (Remarks to the Author):

The authors did a very good job addressing most comments of the reviewers in the revision. I only have one remaining point that the authors need to consider before the publication:

As reviewer 2 also pointed out, in the context of this study it is very important to assess the spread of injection to rule out (or quantify) the leakage to neighboring regions. The authors only partially addressed this point, but I think the paper will benefit from more systematic analysis of the overlaps, and some clarifications.

"The volume of injections was defined as all the voxels with values greater than 50% of the maximum value of Gd brightness on T1w images within a radius of 2.8 mm around the injection site." - could this procedure be illustrated, to see what are the the resulting injection borders and the corresponding estimate of the injection volume? This can be added to Figures S14-17, or as additional figures. The order of S14-17 is confused in the Supplement. I would also crop the relevant hemisphere without midline, to zoom in to the relevant parts of the scan, and to avoid image saturation by very bright intensity in S17.

>> We appreciate the reviewer's excellent suggestions. We have added the injection borders (indicated by the red lines) to Figures S14-17 (S16-17 in the revision, we have changed the order of supplementary figures to reflect their narrative in the manuscript) and have corrected the order of these figures in the revised supplementary figures section. As the reviewer suggested, we have cropped the images to show the injected hemispheres only and zoomed in the injection sites.

Moreover, when we added the injection borders, we noticed a typo. Initially, we selected a relatively larger radius (2.8 mm) for defining the mask of the Gd spread. Later, we found that the radius of 2.5 mm was good enough to cover the Gd spread. We have corrected this typo in the revision. Noted that in Sadagopan et al., 2017, after 5 μ L of muscimol infusion (2.7~ 3.75 μ L of

muscimol infusion in the present study), the average volume of the injection corresponded to a ~2 mm-radius sphere. The size of the mask of the Gd spread does not affect our major results. The Gd results from a 2.8 mm mask were very similar to those from a 2.5 mm mask. In particular, even with a 2.8 mm mask, leakages to neighboring non-targets were still very limited (zero or approaching to zero). Please see below Table R1.

Table R1. Volumetric overlap between the inactivation injections and non-targeted neighboring ROIs under two different radius (2.5 mm vs 2.8 mm) of the mask of the Gd spread. Only the results from F/L alone injections were shown.

	Inactivation Site	2.8 mm-radius sphere ROI Overlap (%) Mean±SD Range	2.5 mm-radius sphere ROI Overlap (%) Mean±SD Range
Monkey C	LAF	0.45±0.39 (0~0.67)	0
	LAL	NA	NA
	RAF	NA	NA
	RAL	NA	NA
	LMF	NA	NA
	LML	0	0
	RMF	0	0
	RML	0	0
Monkey D	LAF	0	0
	LAL	0	0
	RAF	NA	NA
	RAL	3.50	1.02
	LMF	7.17±1.93 (3.77~8.43)	4.08±1.58 (1.26~4.91)
	LML	15.61	13.99
	RMF	NA	NA
	RML	0.41±0.58 (0~1.07)	0.26±0.44 (0~0.77)

The figures S14-17 are not clear: e.g. what does LML inactivation mean in the context of "From a MF&ML session"? - i.e. why not both are shown to be inactivated? In other words, I expected to see Gd only in MF in MF session, only in ML in ML session, and in both ROIs in MF&ML session.

>> We have updated figures S14-17 (S16-17 in the revision) as the reviewer suggested. In the revised Figures S14-17, examples of Gd in AF/AL/AF&AL as well as MF/ML/MF&ML inactivations were collected from corresponding sessions.

"The borders of targeted ROIs are indicated by the blue lines, while the borders of nearby non-targeted ROIs are indicated by the green lines." - but in some cases, it appears that there is as

much Gd within green ROI as within blue ROI (e.g. S15 LML inactivation middle sections, S16 RAL inactivation left section).

>> These two examples were from combined injection sessions (i.e., S15 LML from a MF&ML session and S16 RAL from an AF&AL session). Therefore, Gd also presented within green ROIs. We have re-selected/organized examples as the reviewer suggested in the above comment to avoid such confusion.

[PREVIOUS COMMENT – In the supplement, please show example session scans with Gd, for each face patch, and analyze the spread of Gd across sessions (e.g. using across-session heatmaps).]

RESPONSE >> We have added some example sessions with Gd to show the spread of Gd (Figures S14-17). Since the number of sessions under different inactivation sites was limited, the value of heatmaps presenting whether and how many times one voxel may be inactivated by muscimol could not reflect the spread of Gd well. Instead, we calculated volumetric overlaps between the inactivation injections and face patches for each session using similar methods as that of Sadagopan et al., (2017). We have added these results in Tables S4&5.

I still think the probability maps can easily be done even across varying number of sessions and is a straightforward way to assess both the extent and reproducibility of the injections. The quantification along the lines of Tables S4 and S5 is also OK - but then it needs some indication about variance across sessions - i.e. mean, SD and range of percent overlap of injection and ROI volumes.

>> We appreciate the reviewer's suggestion. We have added mean, SD and range in Table S4 and S5 in the revised supplementary tables. Moreover, we added the probability maps as the reviewer suggested. Since the number of sessions under different inactivation sites was limited, instead of showing the number of sessions covering on one voxel, the probability maps of Gd spread were calculated by averaging percentage intensity (divided by the maximum value of Gd brightness) across all the sessions. The minimum probability is 50% (the threshold for defining the Gd spread) divided by the number of sessions. Please see the updated figures S16-19.

Importantly, I am missing in the Tables S4 and S5 the data on inadvertent (due to leakage) spread of Gd to adjacent ROIs, apparent in the figures. For instance, MF Inactivation row only shows the overlap with MF ROI while the columns related to ML are left empty. Please report the values for the adjacent ROIs (and if they are mostly low, this would be a good argument against the diffusion leakage concern!).

>> We have added the information for adjacent ROIs into the revised Table S4 and S5 as the reviewer suggested. As shown in Tables S4 and S5, the numbers in the non-targeted ROIs were very low, even zero.

Table S4 is presumably only showing data on middle patches, but the legend says "Volumetric overlap between the inactivation injections in the middle face patches and the __four__ IT face patches" - and likewise, S5 is concerned with anterior patches but says "Volumetric overlap between the inactivation injections in the anterior face patches and the __four__ IT face patches".

>> We thank the reviewer for noting this discrepancy. As the anterior and middle face patches

are far apart, no Gd spread were found in the remote sites. Therefore, we have corrected the information in the revised Table S4 and S5 to the following: “Volumetric overlap between anterior/middle face patches and their inactivation injections.”

Minor:

I was wondering if the same concentration of Gd (1:1200) that was used for chamber grid localization was also used for the intracortical injection, and whether the 1:1200 means the 1 part of the Magnevist solution (0.5 mmol/mL?) and 1200 parts of saline, or this diluted proportion (1:1200) also accounts for the dilution in the Magnevist bottle?

>> The solution we used for filling the grid was 1:1200 (v:v dilution, 1 part Magnevist to 1200 parts saline). The concentration for the intracortical injections was 5 mM (AsthaGiri et al., 2011). We have made this information clearer in the revised manuscript.

Reference:

AsthaGiri, A.R., Walbridge, S., Heiss, J.D. & Lonser, R.R. Effect of concentration on the accuracy of convective imaging distribution of a gadolinium-based surrogate tracer Laboratory investigation. *J Neurosurg* **115**, 467-473 (2011).

Again, congratulations on the great work, I look forward to see it published.

>> Thank you very much for your insightful and constructive comments and suggestions.

REVIEWERS' COMMENTS

Reviewer #1 (Remarks to the Author):

The manuscript is almost ready for publication except for the following very minor points.

L. 234. This is the first location where “face selectivity index” is mentioned. But the relevant equation (Eq. 1) is first introduced a bit later (L. 256) and defined afterward (L. 266). These should be moved to the first mentioned location.

Fig. 8. "MF" in the *third* plot from the left, in the bottom row should be "AL".

Reviewer #3 (Remarks to the Author):

The authors have addressed my remaining concerns. I think the manuscript is now ready for the publication.

REVIEWER COMMENTS

Reviewer #1 (Remarks to the Author):

The manuscript is almost ready for publication except for the following very minor points.

>> Thank you very much for your insightful and constructive comments and suggestions.

L. 234. This is the first location where “face selectivity index” is mentioned. But the relevant equation (Eq. 1) is first introduced a bit later (L. 256) and defined afterward (L. 266). These should be moved to the first mentioned location.

>> We agree with the reviewer. We have moved the Eq.1 close to the first mentioned location in the revision.

Fig. 8. "MF" in the *third* plot from the left, in the bottom row should be "AL".

>> Thanks for alerting us to this matter. We have corrected it in the previous revision. However, we uploaded the wrong one in the previous submission. Very sorry. We have made sure that we have corrected it in this revision.